# An Error Analysis of Deep Density-Ratio Estimation with Bregman Divergence

## Abstract

We establish non-asymptotic error bounds for a nonparametric density-ratio estimator using deep neural networks with the Bregman divergence. We also show that the deep density-ratio estimator can mitigate the curse of dimensionality when the data is supported on an approximate low-dimensional manifold. Our error bounds are optimal in the minimax sense and the pre-factors in our error bounds depend on the dimensionality of the data polynomially. We apply our results to investigate the convergence properties of the telescoping density-ratio estimator (Rhodes et al., 2020) and provide sufficient conditions under which it has a smaller upper error bound than a single-ratio estimator.

## 1 Introduction

Density-ratio estimation is of key importance in various statistical and machine learning problems (Sugiyama et al., 2012b; Kato & Teshima, 2021). There is a vast literature on density-ratio estimation due to its wide range of applications, such as discriminative analysis (Silverman, 1978; Cox & Ferry, 1991), covariate shift adaptation (Sugiyama et al., 2008; Tsuboi et al., 2009), two-sample testing (Qin, 1998; Sugiyama et al., 2011), energy-based modelling (Gutmann & Hyvärinen, 2012; Ceylan & Gutmann, 2018), generative learning (Goodfellow et al., 2014; Nowozin et al., 2016), and mutual information estimation (Moustakides & Basioti, 2019; Rhodes et al., 2020), among others.

Let $Z_q$ and $Z_p \in \mathcal{Z} = [0,1]^d$ be two random vectors with probability density functions $q^*$ and $p^*$, respectively. Given independent and identically distributed (i.i.d) samples $\{Z_{q,i}\}_{i=1}^{n_q}$ from $q^*$ and $\{Z_{p,j}\}_{j=1}^{n_p}$ from $p^*$, a basic problem is to estimate the density ratio

$$R^*(z) = q^*(z)/p^*(z), z \in \mathcal{Z}.$$

A naive estimator of $R^*$ is $\hat{q}/\hat{p}$, where $\hat{q}$ and $\hat{p}$ are the density estimators of $q^*$ and $p^*$, respectively. However, such an estimator can be highly unstable. Moreover, density estimation itself is a difficult problem, especially in the high-dimensional settings. For example, kernel density estimators (Rosenblatt, 1956; Parzen, 1962) works well when $d \leq 3$, but deteriorate dramatically as $d$ increases. To avoid density estimation, various methods have been proposed to estimate the density ratio $R^*$ directly, including the density matching approach (Sugiyama et al., 2008; Tsuboi et al., 2009; Yamada & Sugiyama, 2009; Nguyen et al., 2010; Yamada et al., 2010), the moment matching approach (Qin, 1998; Gretton et al., 2009; Kanamori et al., 2012b), the density-ratio fitting approach (Kanamori et al., 2009, 2012a), and the unified density-ratio matching approach under Bregman divergence framework (Sugiyama et al., 2012a). Impressive empirical successes of using deep neural networks in density-ratio estimation have been reported in some recent works (Moustakides & Basioti, 2019; Rhodes et al., 2020). Moreover, Kato & Teshima (2021) studied the convergence properties of deep density-ratio estimation under a modified Bregman divergence criterion.

In this paper, we study deep density-ratio estimators with the Bregman divergence as the criterion. We apply our results to construct an estimator for statistical inference for the Kullback-Liebler divergence. We also study the theoretical properties of the telescoping density-ratio estimator (Rhodes et al., 2020) based on our results.

Our contributions are as follows:

1. We establish non-asymptotic error bounds for the density-ratio estimator using deep neural networks under the Bregman divergence (BD, Bregman, 1967), and provide a neural network architecture for the estimator to achieve minimax optimal rate $O_p\left(n^{-2\beta/(d+2\beta)}\right)$, where $n = \min\{n_q, n_p\}$ and $\beta$ is a smoothness parameter of the logarithmic density-ratio function; see Subsection 3.2 for details;

2. We show that deep density-ratio estimator with the Bregman divergence criterion is able to mitigate the curse of dimensionality when the data is supported on an approximate low-dimensional manifold; see Subsection 3.3;

3. We apply our results to study the convergence properties of the telescoping density-ratio estimator (Rhodes et al., 2020) and demonstrate its advantages over single-ratio estimators under certain conditions.

**Notation.** Let $n = \min\{n_q, n_p\}$ be the smaller sample size between the two samples $\{Z_{q,i}\}_{i=1}^{n_q}$ and $\{Z_{p,j}\}_{j=1}^{n_p}$. In addition, $\|\cdot\|_\infty$ denotes the sup-norm on some specific domain, and $C, C_0$ are generic constants that may vary from place to place. For any measurable function $f$, we denote $\|f\|_{\max} := \max\{\|f\|_p, \|f\|_q\}$ and $\|f\|_{n_p,n_q} = \max\{\|f\|_{p,n_p}, \|f\|_{q,n_q}\}$, where $\|f\|_I^2 = E_{I^*} f^2(Z)$ and $\|f\|_{I,n_I}^2 = E_{n_I} f^2(Z) = (1/n_I) \sum_{t=1}^{n_I} f^2(Z_{I,t}), I = p, q$.

## 2 Density-ratio estimation

In this section, we first present the density-ratio estimation problem using the Bregman divergence (BD, Bregman, 1967) and then describe the structure of the deep neural networks to be used in density-ratio estimation.

Let $\psi : \mathbb{R} \to \mathbb{R}$ be a first-order continuously differentiable and strictly convex function. Define

$$\Delta_\psi(x,y) = \psi(x) - \psi(y) - \psi'(y)(x-y),$$

where $\psi'$ is the derivative of $\psi$. Then, the convexity of $\psi$ implies that $\Delta_\psi(x,y) \geq 0$ and the equality holds if and only if $x = y$. It follows that $E_{p^*}\Delta_\psi(R^*(Z), R(Z)) \geq 0$ and the equality holds if and only if $R = R^*$. Therefore, the target density-ratio $R^* = q^*/p^*$ can be characterized as a minimizer:

$$R^* \in \operatorname*{arg\,min}_{R \text{ nonnegative and measurable}} E_{p^*}\Delta_\psi(R^*(Z), R(Z)).$$

We verify in the appendix that

$$\begin{aligned} &E_{p^*}\Delta_\psi(R^*(Z), R(Z)) \\ &= E_{p^*}[\psi'(R(Z))R(Z) - \psi(R(Z))] - E_{q^*}[\psi'(R(Z))] + E_{p^*}[\psi(R^*(Z))] \end{aligned} \tag{1}$$

Since the last term on the right side in (1) $E_{p^*}[\psi(R^*(Z))]$ is independent of $R$, we have

$$R^* \in \operatorname*{arg\,min}_{R \text{ nonnegative and measurable}} E_{p^*}[\psi'(R(Z))R(Z) - \psi(R(Z))] - E_{q^*}[\psi'(R(Z))]. \tag{2}$$

Hence, for any measurable function $R : \mathcal{Z} \to \mathbb{R}$, the BD score induced by $\psi$ for estimating the target density-ratio $R^* = q^*/p^*$ is

$$\mathcal{B}_\psi(R) = E_{p^*}[\psi'(R(Z))R(Z) - \psi(R(Z))] - E_{q^*}[\psi'(R(Z))], \tag{3}$$

where $\psi'$ is the derivative of $\psi$ (Sugiyama et al., 2012a,b). Then, $R^*$ is the minimizer of $\mathcal{B}_\psi(R)$ over all nonnegative measurable functions.

Because a density ratio is always nonnegative, a nonnegative constraint needs to be considered when defining the density ratio as a minimizer, as in (2). This makes the minimization problem more difficult to solve. To avoid the non-negative constraint of the density ratio, we first consider the

log-density ratio $D^* := \log R^*$. Then the nonnegativity constraint is no longer needed and by (2), we have

$$D^* \in \underset{D \text{ measurable}}{\arg\min} \ \mathcal{B}_\psi(\exp(D)).$$

In practice, the estimation of $R^*$ can be based on an empirical version of $\mathcal{B}_\psi$ when random samples from $p^*$ and $q^*$ are available. Suppose we have samples $\{Z_{q,i}\}_{i=1}^{n_q}$ i.i.d. $q^*$ and $\{Z_{p,j}\}_{j=1}^{n_p}$ i.i.d. $p^*$. We estimate $D^*$ by

$$\widehat{D} \in \underset{D \in \mathcal{F}_n}{\arg\min} \widehat{\mathcal{B}}_\psi(e^D), \tag{4}$$

where $\mathcal{F}_n$ is a class of neural network functions and $\widehat{\mathcal{B}}_\psi(e^D)$ is an empirical version of $\mathcal{B}_\psi(e^D)$ defined in (3), which can be written as

$$\widehat{\mathcal{B}}_\psi(e^D) = \frac{1}{n_p} \sum_{j=1}^{n_p} \mathcal{L}_1(D(Z_{p,j})) + \frac{1}{n_q} \sum_{i=1}^{n_q} \mathcal{L}_2(D(Z_{q,i})),$$

where

$$\mathcal{L}_1(t) = \psi'(e^t)e^t - \psi(e^t) \text{ and } \mathcal{L}_2(t) = -\psi'(e^t). \tag{5}$$

The density-ratio estimator is $\widehat{R} = \exp(\widehat{D})$.

We take the function class $\mathcal{F}_n$ to be $\mathcal{F}_{M,\mathcal{D},\mathcal{W},\mathcal{U},\mathcal{S}}$, a class of ReLU activated feedforward neural networks (FNNs) $f_{\boldsymbol{\theta}} : \mathbb{R}^d \to \mathbb{R}$ with parameter $\boldsymbol{\theta}$, depth $\mathcal{D}$, width $\mathcal{W}$, size $\mathcal{S}$, number of neurons $\mathcal{U}$. We require that $\|f_{\boldsymbol{\theta}}\|_\infty \leq M$ for some $0 \leq M \leq \infty$. There are $\mathcal{D}$ hidden layers and $(\mathcal{D} + 1)$ layers in total. The width $\mathcal{W}$ is the maximum width of the hidden layers; the number of neurons $\mathcal{U}$ is defined as the number of neurons of $f_{\boldsymbol{\theta}}$; the size $\mathcal{S}$ is the total number of parameters in the network. Note that $\mathcal{D}, \mathcal{W}, \mathcal{U}, \mathcal{S}$ may depend on $n$, but we suppress the dependence for notational simplicity. We write $\mathcal{F}_{M,\mathcal{D},\mathcal{W},\mathcal{U},\mathcal{S}}$ as $\mathcal{F}_{\text{FNN}}$ for brevity.

# 3 Theoretical results

In this section, we first study the error bounds for the deep logarithmic density-ratio estimator. The bounds for the density-ratio estimator follows directly based on the properties of the exponential function. We also show that deep density-ratio estimator can mitigate the curse of dimensionality when data is supported on an approximate low-dimensional manifold.

## 3.1 General error bounds

To state our assumptions and results, we need the definitions of $\mu$-smoothness, $\sigma$-strong convexity and pseudo dimension.

A function $f : \mathbb{R} \to \mathbb{R}$ is said to be $\mu$-smooth over a set $\mathcal{A} \subseteq \mathbb{R}$ if it is differentiable over $\mathcal{A}$ and its first-order derivative $f'$ satisfies

$$|f'(x) - f'(y)| \leq \mu|x - y|, \ \forall \ x, y \in \mathcal{A}, \tag{6}$$

where $0 \leq \mu < \infty$. The constant $\mu$ is called the smoothness parameter.

A function $f : \mathbb{R} \to \mathbb{R}$ is called $\sigma$-strongly convex if the domain $\text{dom}(f)$ of $f$ is convex and for any $x, y \in \text{dom}(f)$ and $\lambda \in [0, 1]$, $f$ satisfies

$$f(\lambda x + (1 - \lambda)y) \leq \lambda f(x) + (1 - \lambda)f(y) - \frac{\sigma}{2}\lambda(1 - \lambda)(x - y)^2, \tag{7}$$

where $0 \leq \sigma < \infty$. The constant $\sigma$ is called the strong convexity (SC) parameter.

For a function class $\mathcal{F}$, its pseudo dimension denoted by $\text{Pdim}(\mathcal{F})$, is the largest integer $B$ satisfying that there exists $(x_1, x_2, \ldots, x_B, y_1, y_2, \ldots, y_B) \in \mathcal{Z}^B \times \mathbb{R}^B$ such that for any $(r_1, r_2, \ldots, r_B) \in \{0, 1\}^B$, there exists an $f \in \mathcal{F}$ satisfying for any $i \in \{1, 2, \ldots, B\} : f(x_i) > y_i \Leftrightarrow r_i = 1$ (Anthony & Bartlett, 1999; Bartlett et al., 2019).

Table 1: Commonly-used Loss Functions $\psi$

| Name | $\psi(c)$ | Domain | Smooth Parameter $\mu$ | SC Parameter $\sigma$ |
|------|-----------|--------|------------------------|-----------------------|
| LS | $(c-1)^2$ | $\mathbb{R}$ | 2 | 2 |
| LR | $c\log c - (c+1)\log(c+1)$ | $[a,b]\ (-1 \le a \le b)$ | $\frac{1}{a(a+1)}$ | $\frac{1}{b(b+1)}$ |
| KL | $c\log c - c$ | $[a,b]\ (0 \le a \le b)$ | $\frac{1}{a}$ | $\frac{1}{b}$ |

**Remark 1.** *For any measurable function class $\mathcal{F}$, by the definition of VC dimension, $\mathrm{VCdim}(\mathcal{F}) \le \mathrm{Pdim}(\mathcal{F})$. If $\mathcal{F}$ is the class of functions generated by ReLU FNNs, it follows from Theorem 14.1 of Anthony & Bartlett (1999) that $\mathrm{Pdim}(\mathcal{F}) \le \mathrm{VCdim}(\mathcal{F})$. Hence, for the function class $\mathcal{F}$ generated by ReLU FNNs, $\mathrm{Pdim}(\mathcal{F}) = \mathrm{VCdim}(\mathcal{F})$.*

We make the following assumptions.

**Assumption 1.** *The function $\psi$ is $\mu$-smooth & $\sigma$-strongly convex, that is, it satisfies (6) and (7).*

Some commonly-used $\psi$'s satisfy Assumption 1; see Table 1 for some examples.

**Assumption 2.** *There exists a constant $0 < M < \infty$ such that $\|D^*\|_\infty \le M, \|D\|_\infty \le M$ for every $D \in \mathcal{F}_{\mathrm{FNN}}$.*

Assumption 2 assumes that the target density ratio is bounded. Such an assumption is often made in nonparametric statistics for avoiding technical difficulties associated with dealing with unbounded functions. We will partially relax this assumption below. The finite $M$ in Assumption 2 can be relaxed to $M = \mathcal{O}(\log\log n)$ at a small price of an additional logarithm term in the error bounds. The boundedness of a network can be achieved by clipping operation. For example, let $T_M(t) = -MI\{t < -M\} + tI\{-M \le t \le M\} + MI\{t > M\}$ be the truncation function taking values in $[-M, M]$, then $T(t) = \sigma(t) - \sigma(\sigma(t) - M) - \{\sigma(-t) - \sigma(\sigma(-t) - M)\}$ can be computed by a ReLU network with depth 2 and width 4. Hence, through network concatenation, we can construct some bounded ReLU FNNs and such a boundedness assumption can be satisfied.

Define the best in class approximation of $D^*$ in $\mathcal{F}_{\mathrm{FNN}}$ as $D_{\mathrm{NN}} \in \arg\min_{D \in \mathcal{F}_{\mathrm{FNN}}} \|D - D^*\|_{\max}$. Denote

$$\xi_n = \sqrt{\frac{\mathrm{Pdim}(\mathcal{F}_{\mathrm{FNN}})\log n}{n}}. \tag{8}$$

**Theorem 1.** *Suppose Assumptions 1-2 are satisfied. When $n \ge \mathrm{Pdim}(\mathcal{F}_{\mathrm{FNN}})$, there exists a constant $C$ depending on $(\mu, \sigma, M)$ such that for any $\gamma > 0$, with probability at least $1 - \exp(-\gamma)$,*

$$\|\widehat{D} - D^*\|_{\max} \le C\left(\xi_n + \|D_{\mathrm{NN}} - D^*\|_{\max} + \sqrt{\frac{\gamma}{n}}\right),$$

*and*

$$\|\widehat{D} - D^*\|_{n_p, n_q} \le 2C\left(\xi_n + \|D_{\mathrm{NN}} - D^*\|_{\max} + \sqrt{\frac{\gamma}{n}}\right).$$

We have the following corollary for the expected error.

**Corollary 1.** *Under the conditions of Theorem 1, there exists a constant $C_0$ depending only on $(\mu, \sigma, M)$, such that*

$$E_{p^*, q^*}\|\widehat{D} - D^*\|_{\max}^2 \le C_0\left(\xi_n^2 + \|D_{\mathrm{NN}} - D^*\|_{\max}^2\right),$$

*and*

$$E_{p^*, q^*}\|\widehat{D} - D^*\|_{n_p, n_q}^2 \le 2C_0\left(\xi_n^2 + \|D_{\mathrm{NN}} - D^*\|_{\max}^2\right).$$

The above results are obtained under the boundedness Assumption 2. While such an assumption is often made in the error analysis of nonparametric procedures, it is somewhat restrictive in density-ratio estimation problems. For example, this assumption may not be satisfied in the presence of the density-chasm problem, i.e., the gap between two densities is large (Rhodes et al., 2020). We establish an error bound result with the following partially relaxed assumption.

**Assumption 3.** *There exists a constant $0 < M < \infty$ such that $D^*(z) \geq -M$ for every $z \in \mathcal{Z}$ and $\|D\|_\infty \leq M$ for every $D \in \mathcal{F}_{\mathrm{FNN}}$.*

This assumption does not require the target log-density ratio $D^*$ to be bounded above. Denote truncated versions of $D^*$ and $R^*$ by

$$D_M^*(z) = D^*(z)\mathbf{1}\{D^*(z) \leq M\} + M\mathbf{1}\{D^*(z) \geq M\},$$
$$R_M^*(z) = R^*(z)\mathbf{1}\{R^*(z) \leq e^M\} + e^M\mathbf{1}\{R^*(z) \geq e^M\},$$

where $0 < M < \infty$ and $\mathbf{1}\{\cdot\}$ is the indicator function. We establish a non-asymptotic error bound involving the truncation error.

**Theorem 2.** *Suppose Assumptions 1 and 3 hold. When $n > \mathrm{Pdim}(\mathcal{F}_{\mathrm{FNN}})$, there exists two constants $C$ depending only on $(\mu, \sigma, M)$ and $C_0$ depending only on $(\mu, \sigma)$, such that*

$$E_{p^*, q^*}\|\widehat{D} - D^*\|_p^2 \leq C_0 e^{2M}\|R^* - R_M^*\|_p^2 + C\left(\xi_n + \inf_{D \in \mathcal{F}_{\mathrm{FNN}}}\|D - D_M^*\|_p^2\right),$$

*where $\xi_n$ is defined in (8).*

The term $\|R^* - R_M^*\|_p^2$ is the truncation error for an unbounded $R^*$ and the unboundedness also leads to the term $\xi_n = [\mathrm{Pdim}(\mathcal{F}_{\mathrm{FNN}})(\log n)/n]^{1/2}$ in the error bound, which is greater than $\xi_n^2$ in the bounded case. However, because no boundedness assumption is needed in this theorem, we can apply it to study the convergence properties of the telescoping density-ratio estimator of Rhodes et al. (2020) in Section 4 below.

## 3.2 Non-asymptotic error bounds

By Corollary 1, it suffices to bound the estimation error $\mathrm{Pdim}(\mathcal{F}_{\mathrm{FNN}})\log n/n$ and the approximation error $\|D_{\mathrm{NN}} - D^*\|_{\max}^2$. It follows from Theorem 6 in Bartlett et al. (2019) that, for $\mathcal{F}_{\mathrm{FNN}} = \mathcal{F}_{M, \mathcal{D}, \mathcal{W}, \mathcal{U}, \mathcal{S}}$, there exists a universal constant $C_2$ such that $\mathrm{Pdim}(\mathcal{F}_{\mathrm{FNN}}) \leq C_2 \mathcal{S}\mathcal{D}\log \mathcal{S}$. To control the approximation error $\|D_{\mathrm{NN}} - D^*\|_{\max}^2$, we assume that $D^*$ belongs to the Hölder class $\mathcal{H}^\beta([0,1]^d, M)$ with $\beta = k + a$ where $k \in \mathbb{N}^+$ and $a \in (0,1]$, where $\mathbb{N}^+$ is the set of positive integers.

**Definition 1** (Hölder class). *A Hölder class $\mathcal{H}^\beta([0,1]^d, M)$ with $\beta = k + a$ where $k \in \mathbb{N}^+$ and $a \in (0,1]$ consists of function $f : [0,1]^d \to \mathbb{R}$ satisfying*

$$\max_{\|\boldsymbol{\alpha}\|_1 \leq k}\|\partial^{\boldsymbol{\alpha}} f\|_\infty, \max_{\|\boldsymbol{\alpha}\|_1 = k}\max_{x \neq y}\frac{|\partial^{\boldsymbol{\alpha}} f(x) - \partial^{\boldsymbol{\alpha}} f(y)|}{\|x - y\|_2^a} \leq M,$$

*where $\|\boldsymbol{\alpha}\|_1 = \sum_{i=1}^d \alpha_i$ and $\partial^{\boldsymbol{\alpha}} = \partial^{\alpha_1}\partial^{\alpha_2}\cdots\partial^{\alpha_d}$ for $\boldsymbol{\alpha} = (\alpha_1, \alpha_2, \ldots, \alpha_d) \in \mathbb{N}^{+d}$.*

We use Theorem 3.3 of Jiao et al. (2021) to control the approximation error $\|D_{\mathrm{NN}} - D^*\|_{\max}^2$. For convenience, we include this result in the following lemma.

We specify the width $\mathcal{W}$ and depth $\mathcal{D}$ as follows. For any $K, L \in \mathbb{N}^+$,

$$\mathcal{W} = 38(\lfloor\beta\rfloor + 1)^2 d^{\lfloor\beta\rfloor + 1} L\lceil\log_2(8L)\rceil, \mathcal{D} = 21(\lfloor\beta\rfloor + 1)^2 K\lceil\log_2(8K)\rceil, \tag{9}$$

where $\lceil a \rceil$ is the smallest integer no less than $a$.

**Lemma 1** (Approximation error). *Assume $f \in \mathcal{H}^\beta([0,1]^d, M)$ with $\beta = k + a$ where $k \in \mathbb{N}^+$ and $a \in (0,1]$. Then there exists a function $\phi_0$ implemented by a ReLU network with width $\mathcal{W}$ and depth $\mathcal{D}$ specified in (9) such that*

$$\sup_{x \in [0,1]^d \setminus H_{B,\delta}}|f - \phi_0| \leq 18 M C_\beta (KL)^{-\frac{2\beta}{d}},$$

*where $C_\beta = (\lfloor\beta\rfloor + 1)^2 d^{\lfloor\beta\rfloor + (\beta\vee 1)/2}$, $H_{B,\delta} = \cup_{i=1}^d\{x = [x_1, \ldots, x_d] : x_i \in \cup_{b=1}^{B-1}(b/B - \delta, b/B)\}$ for $B = \lceil(KL)^{2/d}\rceil, \delta \in (0, 1/(3B)]$ and $a \vee b = \max(a, b)$.*

*Furthermore, if $\mathcal{W} = 38(\lfloor\beta\rfloor + 1)^2 d^{\lfloor\beta\rfloor + 1}3^d L\lceil\log_2(8L)\rceil, \mathcal{D} = 21(\lfloor\beta\rfloor + 1)^2 K\lceil\log_2(8K)\rceil + 2d$, then*

$$\sup_{x \in [0,1]^d}|f - \phi_0| \leq 19 M C_\beta (KL)^{-\frac{2\beta}{d}}.$$

*In this uniform approximation result, the width $\mathcal{W}$ is required to depend on $d$ exponentially.*

The following theorem gives an error bound for $\widehat{D}$.

**Theorem 3** (Non-asymptotic error bound for $\widehat{D}$). *Suppose that Assumptions 1-2 are satisfied, $D^* \in \mathcal{H}^\beta([0,1]^d, M)$ with $\beta = k + a$ where $k \in \mathbb{N}^+$ and $a \in (0,1]$, and $\mathcal{F}_{\text{FNN}}$ is the function class of ReLU DNNs with width $\mathcal{W}$ and depth $\mathcal{D}$ specified in (9). Then, for $M \geq 1$ and $n \geq \text{Pdim}(\mathcal{F}_{\text{FNN}})$, we have*

$$E_{p^*,q^*}\|\widehat{D} - D^*\|_{\max}^2 \leq C\left(\xi_n^2 + C_1(KL)^{-\frac{4\beta}{d}}\right),$$

*where $C_1 = (\lfloor\beta\rfloor + 1)^4 d^{2\lfloor\beta\rfloor + (\beta\vee 1)}$ and the constant $C$ depends only on $(\mu, \sigma, M)$.*

*Furthermore, if*

$$\mathcal{W} = 114(\lfloor\beta\rfloor + 1)^2 d^{\lfloor\beta\rfloor + 1}, \mathcal{D} = 21(\lfloor\beta\rfloor + 1)^2 \left\lceil n^{\frac{d}{2(d+2\beta)}} \log_2\left(8n^{\frac{d}{2(d+2\beta)}}\right)\right\rceil,$$

*then*

$$E_{p^*,q^*}\|\widehat{D} - D^*\|_{\max}^2 \leq C_0(\lfloor\beta\rfloor + 1)^9 d^{2\lfloor\beta\rfloor + (\beta\vee 3)} n^{-\frac{2\beta}{d+2\beta}}, \tag{10}$$

*where the constant $C_0$ depends only on $(\mu, \sigma, M)$.*

The convergence rate in (10) is optimal. This can be seen by considering a density estimation problem with i.i.d observations $\{Z_{q,i}^{(1)}\}_{i=1}^{m_q}$ from an underlying unknown density $q_1$ on $[0,1]^d$. To estimate $q_1$, we sample referencing observations $\{Z_{p,j}^{(1)}\}_{j=1}^{m_p}$ with $m_p \geq m_q$, from a uniform distribution $\text{Unif}([0,1]^d)$ whose density $p_1 \equiv 1$. Thus, estimating the density ratio $q_1/p_1$ is equivalent to estimating $q_1$. According to (4), we obtain the estimator $\hat{q}_1$ of $q_1$. If $\log q_1 \in \mathcal{H}^\beta([0,1]^d, M)$ where $\beta = k + a$ with $k \in \mathbb{N}^+$ and $a \in (0,1]$, a neural estimator based on the network structure specified in Theorem 3 satisfies

$$E_{p_1,q_1}\|\hat{q}_1 - q_1\|_{\max}^2 \leq C_0(\lfloor\beta\rfloor + 1)^9 d^{2\lfloor\beta\rfloor + (\beta\vee 3)} m_q^{-\frac{2\beta}{d+2\beta}}. \tag{11}$$

Tsybakov (2008) showed that for a density belonging to the Hölder function class, the optimal minimax rate of the density estimation is $O_p\left(m_q^{-2\beta/(d+2\beta)}\right)$. Hence, our estimator achieves the optimal minimax rate.

In addition, the existing error bounds usually contain a prefactor depending on the dimension $d$ exponentially, e.g. $2^d$ (Devroye & Lugosi, 1996). Such a prefactor can be very large even for a moderately large $d$, which severely degrades the quality of an error bound. The prefactors in our results depend on $d$ only polynomially and are much smaller than those in the existing bounds.

Under Assumption 2, to derive a nonasymptotic error bound for the log-density ratio estimator $\widehat{R}$, we note that

$$E_{p^*,q^*}\|\widehat{R} - R^*\|_{\max}^2 \leq e^{2M} E_{p^*,q^*}\|\widehat{D} - D^*\|_{\max}^2.$$

Thus a bound for $\widehat{R}$ follows directly from a bound for $\widehat{D}$.

**Remark 2.** *Appendix A.2 contains some examples of $p^*$ and $q^*$ such that $D^* = \log(q^*/p^*) \in \mathcal{H}^\beta([0,1]^d, M)$.*

**Remark 3.** *The hypercube $[0,1]^d$ assumption for the density ratio is made for technical convenience. With an unbounded support, we can bound $\|D_{\text{NN}} - D^*\|_{\max}$ using a truncation technique under some suitable additional assumptions, at a small price of an additional logarithm term in the error bound. Specifically, suppose the pdfs are supported on $\mathbb{R}^d$. In addition to Assumptions 1-2 and the Hölder class assumption in Theorem 3, we need to further assume that $\max\{E_{p^*}I(\|Z\|_\infty \geq \log n), E_{q^*}I(\|Z\|_\infty \geq \log n)\} \leq n^{-\frac{2\beta}{d+2\beta}}$. For $I = p$ or $q$, and any $D \in \mathcal{F}_{\text{FNN}}$, where $\mathcal{F}_{\text{FNN}}$ is the function class of ReLU FNNs with width $\mathcal{W}$ and depth $\mathcal{D}$ specified by*

$$\mathcal{W} = 114(\lfloor\beta\rfloor + 1)^2 d^{\lfloor\beta\rfloor + 1}, \mathcal{D} = 21(\lfloor\beta\rfloor + 1)^2\left\lceil n^{\frac{d}{2(d+2\beta)}} \log_2\left(8n^{\frac{d}{2(d+2\beta)}}\right)\right\rceil,$$

*we have*

$$\|D_{\text{NN}} - D^*\|_{\max}^2 \leq 328M^2(\lfloor\beta\rfloor + 1)^4 d^{2\lfloor\beta\rfloor + (\beta\vee 1)}(2\log n)^{2\lfloor\beta\rfloor} n^{-\frac{2\beta}{d+2\beta}}.$$

*Compared with the upper bound of the approximation error in Theorem 3, when the pdfs are supported on $\mathbb{R}^d$ (unbounded case), we can derive a similar approximation error upper bound with an additional logrithmic factor $(2\log n)^{2\lfloor\beta\rfloor}$. The details are given in Appendix A.3.*

## 3.3 Circumventing the curse of dimensionality

In many modern statistical and machine learning tasks, such as image processing and text analysis, the dimensionality $d$ of the data can be high, which results in a very slow convergence rate even with a large sample size. This is known as the curse of dimensionality. Nonetheless, the data in various applications has been demonstrated to be supported or approximately supported in some subspaces or subsets with low intrinsic dimensionality (Nakada & Imaizumi, 2020). For regression problems, Nakada & Imaizumi (2020) have shown that DNNs can adaptively estimate the regression function through the low-dimensional structure of the data, and the resulting convergence rates no longer depend on the nominal high dimensionality $d$ of the data, but on its low intrinsic dimension.

Motivated by these advancements, we assume that the data is concentrated on an approximate compact Riemannian submanifold $\mathcal{M}$ with the Riemannian dimension $d_{\mathcal{M}} \ll d$.

**Assumption 4.** *The target log-density ratio $D^* \in \mathcal{H}^\beta([0,1]^d, M)$ with $\beta = k + a$ where $k \in \mathbb{N}^+$ and $a \in (0,1]$, and the data from the densities $p^*, q^*$ are concentrated on a set $\mathcal{M}_\rho \subseteq [0,1]^d$ defined as*

$$\mathcal{M}_\rho := \{x \in [0,1]^d : \text{ there exists } y \in \mathcal{M}, \|x - y\|_2 \leq \rho\},$$

*where $\mathcal{M}$ is a compact $d_{\mathcal{M}}$-dimensional Riemannian submanifold and $\rho \in (0,1)$.*

**Theorem 4.** *Suppose Assumptions 1, 2 and 4 hold. Suppose that $D^* \in \mathcal{H}^\beta([0,1]^d, M)$ with $\beta = k + a$, $k \in \mathbb{N}^+$ and $a \in (0,1]$. If $\mathcal{F}_{\text{FNN}}$ is the function class of ReLU FNNs with width and depth*

$$\mathcal{W} = 38(\lfloor\beta\rfloor + 1)^2 d_\delta{}^{\lfloor\beta\rfloor+1} L\lceil\log_2(8L)\rceil, \mathcal{D} = 21(\lfloor\beta\rfloor + 1)^2 K\lceil\log_2(8K)\rceil,$$

*where $K, L \in \mathbb{N}^+$ and $d_\delta = O\left(d_{\mathcal{M}}\log(d/\delta)/\delta^2\right) \ll d$, then when $M \geq 1$, $n > \text{Pdim}(\mathcal{F}_{\text{FNN}})$ and*

$$\rho \leq (\lfloor\beta\rfloor + 1)^2 2^\beta d^{\beta - \frac{1}{2}} d_\delta^{\lfloor\beta\rfloor + (\beta-1/2)\vee(1/2)}(KL)^{-\frac{2\beta}{d_\delta}},$$

*we have*

$$E_{p^*,q^*}\|\widehat{D} - D^*\|_{\max}^2 \leq C(1-\delta)^{-2\beta}\left[\xi_n^2 + C_2(KL)^{-\frac{4\beta}{d_\delta}}\right],$$

*where the constant $C$ only depends on $(\mu, \sigma, M)$, $C_2 = (\lfloor\beta\rfloor + 1)^4(2d)^{2\beta}d_\delta^{3\beta+(\beta\vee1)}$, and $\xi_n$ is defined in (8).*

By Theorem 4, if we set $\mathcal{W} = 114(\lfloor\beta\rfloor + 1)^2 d_\delta^{\lfloor\beta\rfloor+1}, \mathcal{D} = 21(\lfloor\beta\rfloor + 1)^2\lceil n^{\zeta_\delta}\log_2(8n^{\zeta_\delta})\rceil$, with $\zeta_\delta = d_\delta/(2(d_\delta + 2\beta))$, then

$$E_{p^*,q^*}\|\widehat{D} - D^*\|_{\max}^2 \leq C_0 C_3(1-\delta)^{-2\beta}n^{-\frac{2\beta}{d_\delta+2\beta}}, \tag{12}$$

where the constant $C_0$ only depends on $(\mu, \sigma, M)$ and $C_3 = (\lfloor\beta\rfloor + 1)^9\max\{d_\delta^{2\lfloor\beta\rfloor+3}, (2d)^{2\beta}d_\delta^{3\beta+(\beta\vee1)}\}$. The convergence rate $n^{-2\beta/(d_\delta+2\beta)}$ in (12) only depends on $d_\delta \ll d$, instead of the ambient dimension $d$. Therefore, Theorem 4 shows that a low-dimensional Riemannian manifold support assumption can alleviate the curse of dimensionality.

## 4 Error analysis of the telescoping density-ratio estimator

When the difference or the 'gap' between two densities is large, a single-ratio estimation method may perform poorly. This is referred to as the the *density-chasm problem* (Rhodes et al., 2020). To alleviate this problem, Rhodes et al. (2020) proposed an approach called Telescoping density-Ratio Estimation (TRE). This approach first gradually transports samples from $q^*$ to samples from $p^*$, creating a chain of intermediate datasets, then estimates the density ratio between consecutive datasets along this chain. The chained ratios are combined via a telescoping product which yields an estimate of the original density ratio. The experiments conducted by Rhodes et al. (2020) demonstrate that TRE can yield substantial improvements over existing single-ratio methods for mutual information estimation, representation learning and energy-based modelling.

We now provide a theoretical analysis of TRE, which partially explains why TRE performs well. For notational simplicity, suppose $n_p = n_q \equiv n$ below.

For $k = 0, 1, \ldots, K$, Rhodes et al. (2020) constructed a chain of intermediate samples connecting $q^*$ and $p^*$ by setting $Z_{k,i} = (1 - \alpha_k^2)^{1/2} Z_{q,i} + \alpha_k Z_{p,i}$, $i = 1, \ldots, n$, where $0 = \alpha_0 < \alpha_1 < \cdots < \alpha_{K-1} < \alpha_K = 1$, and used these samples to build a TRE.

To simplify the analysis, we use a slightly different chain of intermediate samples as follows. For $k = 0, 1, \ldots, K$, let

$$Z_{k,i} = (1 - \delta_{k,i}) Z_{q,i} + \delta_{k,i} Z_{p,i}, \ i = 1, \ldots, n, \tag{13}$$

where $\delta_{k,i}, i = 1, \ldots, n$, are i.i.d. Bernoulli random variables with success probability $\alpha_k$.

Let $q_k$ be the density of the synthetic data $Z_{k,i}$ constructed this way. We have $q_k(z) = (1 - \alpha_k) q^*(z) + \alpha_k p^*(z), k = 1, \ldots, K - 1$. Therefore, the distribution of the samples from $q_k$ in the chain is a simple mixture of $q^*$ and $p^*$ with the mixing proportions $1 - \alpha_k$ and $\alpha_k$, instead of a more complex convolution of two densities using the construction of Rhodes et al. (2020). As $\alpha_k$ changes from $\alpha_0 = 0$ to $\alpha_K = 1$ over a grid $\{\alpha_0, \alpha_1, \ldots, \alpha_K\} \subset [0, 1]$, the distributions of the samples in the chain move gradually from $q^*$ to $p^*$. Let $q_0 = q^*$ and $q_K \equiv p^*$. Then,

$$R^*(z) = \frac{q^*(z)}{p^*(z)} = \prod_{i=0}^{K-1} R_i^*(z), \ z \in \mathcal{Z}, \tag{14}$$

where $R_i^*(z) = q_i(z)/q_{i+1}(z)$. For $k = 0, 1, \ldots, K - 1$, applying the neural density-ratio estimator with $\{Z_{k,j}\}_{j=1}^{n_k}$ and $\{Z_{k+1,j}\}_{j=1}^{n_{k+1}}$ yields an estimator $\widehat{R}_i$ of $R_i^*$. Then the telescoping density ratio estimator of $R^*$ is $\prod_{i=0}^{K-1} \widehat{R}_i$

We consider the log-density ratio. Let $\widehat{D}_k$ be the neural estimator of $D_k^* \equiv \log(q_k/q_{k+1})$. Based on (14), the telescoping estimator of the log-density ratio $D^* \equiv \log R^*$ is

$$\widehat{D}_{\text{TRE}} = \sum_{k=0}^{K-1} \widehat{D}_k. \tag{15}$$

In what follows, we show that under certain conditions, the telescoping estimator has an improved asymptotic error bound. The intuition is as follows: when $q_k/q_{k+1}$ is bounded or $q_k(z)/q_{k+1}(z) \ll q^*(z)/p^*(z)$ for $z \in \mathcal{Z}$, where $q_k$ and $q_{k+1}$ are the densities of the synthetic data $\{Z_{k,j}\}_{j=1}^{n}$ and $\{Z_{k+1,j}\}_{j=1}^{n}$, respectively, the truncation error for $q_k/q_{k+1}$ vanishes or is far less than that for $q^*/p^*$. This can help the telescoping density-ratio estimator perform better than a single-ratio estimator.

Assume that $q^* \geq c_1$ and $c_1 \leq p^* \leq c_2$, where $0 < c_1, c_2 < \infty$ are two constants. Thus, $D^* = \log(q^*/p^*) \geq \log(c_1/c_2)$. Therefore, Assumption 3 is satisfied. For any finite set $\mathcal{A} \subset \mathbb{R}$, $\max \mathcal{A}$ denotes the maximal value in $\mathcal{A}$. Let $M = \log(c_2/c_1)$ and $M_0$ be a constant satisfying

$$M_0 \geq \max \mathcal{A}_{M,\alpha}^{(2K)}, \tag{16}$$

where

$$\mathcal{A}_{M,\alpha}^{(2K)} = \left\{ M, 1 \right\} \cup \left\{ \log \frac{1 - \alpha_{k-1}}{1 - \alpha_k}, 1 \leq k \leq K - 1 \right\} \cup \left\{ \log \frac{(e^M - 1)\alpha_k + 1}{(e^M - 1)\alpha_{k-1} + 1}, 1 \leq k \leq K - 1 \right\}.$$

Based on Theorem 2, we can establish an asymptotic error bound for the telescoping estimator $\widehat{D}_{\text{TRE}}$ defined in (15), with

$$\widehat{D}_k \in \underset{D \in \mathcal{F}_{\text{FNN}}^0}{\arg\min} \hat{\mathcal{B}}_\psi^k(e^D),$$

where

$$\hat{\mathcal{B}}_\psi^k(e^D) = \frac{1}{n} \sum_{i=1}^n \mathcal{L}_1(D(Z_{k+1,i})) + \frac{1}{n} \sum_{i=1}^n \mathcal{L}_2(D(Z_{k,i})),$$

where $\mathcal{L}_1$ and $\mathcal{L}_2$ are defined in (5) and $\mathcal{F}_{\text{FNN}}^0$ consists of DNNs $D$ with $\|D\|_\infty \leq M_0$. To demonstrate the advantages of the telescoping estimator, we also consider the single-ratio estimator (SRE), $\widehat{D}_{\text{SRE}} \in \arg\min_{D \in \mathcal{F}_{\text{FNN}}^0} \hat{\mathcal{B}}_\psi(e^D)$.

**Proposition 1.** *Assume that $q^* \geq c_1$, $c_1 \leq p^* \leq c_2$, where the constants $0 < c_1 \leq c_2 < \infty$, and the samples $\{Z_{q,i}\}_{i=1}^n$ from $q^*$ and $\{Z_{p,j}\}_{j=1}^n$ from $p^*$ are independent. Then, there exists a constant $C_0(\mu, \sigma, c_1)$ depending only on $(\mu, \sigma, c_1)$ such that for*

$$B_{\mathrm{SRE}} = e^{M_0} C_0(\mu, \sigma, c_1) \|R^* - R_{M_0}^*\|_p,$$

*we have*

$$\limsup_{n\to\infty} E_{p^*,q^*} \|\widehat{D}_{\mathrm{SRE}} - D^*\|_2 \leq B_{\mathrm{SRE}},$$

$$\limsup_{n\to\infty} E_{p^*,q^*} \|\widehat{D}_{\mathrm{TRE}} - D^*\|_2 \leq (1 - \alpha_{K-1}) B_{\mathrm{SRE}},$$

*where $\|f\|_2 = \left[\int_{\mathcal{Z}} f^2(z)dz\right]^{1/2}$ for any square integrable function $f$.*

Proposition 1 shows that for a given sequence $0 = \alpha_0 < \alpha_1 < \cdots < \alpha_{K-1} < \alpha_K = 1$ and a truncation level $M_0$, the upper bound for the asymptotic $L_2$-error of $\widehat{D}_{\mathrm{TRE}}$ is reduced by a factor $(1 - \alpha_{K-1})$ with $0 < 1 - \alpha_{K-1} < 1$. This upper bound can be far less than that of $\widehat{D}_{\mathrm{SRE}}$ when $\alpha_{K-1}$ is close to 1. Therefore, TRE can improve the asymptotic error bound over the bound for the single-ratio method.

It is important to note that there is a tradeoff between the value of $\alpha_{K-1}$ and the truncation level $M_0$ dictated by (16). For instance, with $\alpha_1 \leq \cdots \leq \alpha_{K-2}$ fixed, the closer $\alpha_{K-1}$ is to 1, in view of (16), the larger $M_0$ is. Larger $\alpha_{K-1}$ sharpens the pre-factor $(1 - \alpha_{K-1})$ and larger $M_0$ also improves $\|R^* - R_{M_0}^*\|_p$, but deteriorates the pre-factor $e^{M_0}$.

Proposition 1 is generally not applicable to the original chain of TRE. The difficulty is due to the possibly intensive oscillation of density ratios caused by the convolution form for the density of the sum of two random variables. We illustrate this by a toy example: suppose $Z_q, Z_p$ are i.i.d. uniform random variables on $[0, 1]$. For any $t \in (0, 1/2]$, $(1 - t)Z_q + tZ_p$ has density $q_t^*(z) = \frac{z}{t(1-t)} I\{0 \leq z \leq t\} + \frac{1}{1-t} I\{t < z \leq 1-t\} + \frac{1-z}{t(1-t)} I\{1-t < z \leq 1\}$. In this case, $q^*/q_t^*$ is unbounded and oscillates sharply when $z$ is close to 0 or 1. This makes it hard to estimate $q^*/q_t^*$. However, the chain we used does not have this problem, which may be a good choice in practice.

Additionally, we conduct simulation studies to evaluate the performance of our proposed mixing chain and the original convolution chain; see Table 2 for the results. The simulation settings are given in Appendix A.4. Table 2 shows that, for the models considered in the simulation studies, the proposed mixing chain performs comparably or better compared with the original convolution chain.

Table 2: The MSEs averaged over 10 replications and the corresponding standard errors in parentheses between the telescoping ratio estimate (TRE) of log density-ratio and its true value for the proposed mixing chain (mTRE) and the original convolution chain (cTRE) under different settings, where $n$ is the training data sample size and $K$ is the length of the chain. The bold one is the best in a specific setting among the two estimates.

| Setting | Method | (n,K) | | | |
|---|---|---|---|---|---|
| | | (5000,5) | (5000,10) | (10000,5) | (10000,10) |
| Beta | mTRE(ours) | **0.9850(0.0269)** | **0.8840(0.0180)** | **1.0109(0.0171)** | **0.9299(0.0194)** |
| | cTRE | 1.4670(0.0606) | 1.2935(0.0274) | 1.3674(0.0625) | 1.2850(0.0293) |
| Normal | mTRE(ours) | **2.7426(0.0370)** | 2.8330(0.0450) | **2.7483(0.0367)** | 2.7813(0.0265) |
| | cTRE | 2.7987(0.0586) | **2.7076(0.0293)** | 2.8184(0.0347) | **2.7503(0.0297)** |

# 5 Related work: comparison with the NN-BD estimator

There has been much work on the error analysis of nonparametric density-ratio estimation (Nguyen et al., 2010; Sugiyama et al., 2008; Kanamori et al., 2012a; Yamada et al., 2013). These results show that when the targeted density-ratio belongs to certain function space $\mathcal{H}$, such as RKHS, and thus no approximation error is incurred, their estimators achieve certain nonparametric convergence rate

decided by the complexity of $\mathcal{H}$. Compared to these works, our results consider the approximation error using neural network functions and still achieves the minimax optimality under some mild conditions.

Our work is most related to the paper by Kato & Teshima (2021), who proposed a non-negative Bregman divergence (NN-BD) method to tackle the possible over-fitting problem due to the unboundedness of certain Bregman divergences. We compare our theoretical results with those for the NN-BD estimator of Kato & Teshima (2021). Using the notation in this paper, we restate two conditions required in Kato & Teshima (2021):

(a) Let $\mathcal{F}_{\mathrm{FNN}}^R$ be a class of FNNs with output taking values in $[e^{-M}, e^M]$ for some finite $M > 0$. The target density-ratio $R^* \in \mathcal{F}_{\mathrm{FNN}}^R$. Moreover, for any function in $\mathcal{F}_{\mathrm{FNN}}^R$, its Frobenius norm of the parameter matrix $W_j$ in the $j$th layer is bounded by $\mathcal{B}_j \geq 0$ and the activation functions are 1-Lipschitz positive-homogeneous.

(b) The function $\psi(\cdot)$ is $\sigma$-strongly convex. Let $\ell_1(t) = \psi^*(t)t - \psi(t) + A, \ell_2(t) = -\tilde{\psi}(t), t \in [e^{-M}, e^M]$, where $\psi^*(t) = C_{nn}\{\psi'(t)t - \psi(t)\} + \tilde{\psi}(t)$. Here $\tilde{\psi}(t)$ is a function bounded above, $C_{nn}$ and $A$ are user-selected constants. Suppose $\ell_1(\cdot)$ and $\ell_2(\cdot)$ are Lipschitz functions on $[e^{-M}, e^M]$.

Under these two conditions, Kato & Teshima (2021) rewrote the BD in (3) as

$$\mathcal{B}_\psi(R) = E_p\ell_1(R(Z)) - C_{nn}E_q\ell_1(R(Z)) + E_q\ell_2(R(Z)) + (1 - C_{nn})A, \qquad (17)$$

and proposed the density-ratio estimator $\widehat{R}_{\mathrm{KT}}$ defined as

$$\widehat{R}_{\mathrm{KT}} \in \underset{R \in \mathcal{F}_{\mathrm{FNN}}}{\arg\min} \left\{ \frac{1}{n_q} \sum_{i=1}^{n_q} \ell_2(R(Z_{q,i})) + \left[ \frac{1}{n_p} \sum_{j=1}^{n_p} \ell_1(R(Z_{p,j})) - \frac{C_{nn}}{n_q} \sum_{j=1}^{n_q} \ell_1(R(Z_{q,j})) \right]_+ \right\},$$

where $[a]_+ = \max(0, a)$ for any $a \in \mathbb{R}$. They showed that

$$\|\widehat{R}_{\mathrm{KT}} - R^*\|_p = O_p\left(n^{-1/(2+a)}\right) \qquad (18)$$

for any $0 \leq a \leq 2$.

According to Theorem 1, we have the following corollary for our density-ratio estimator $\widehat{R} = \exp(\widehat{D})$.

**Corollary 2.** *Under Assumption* 1, *when* $n \geq \mathrm{Pdim}(\mathcal{F}_{\mathrm{FNN}}^R)$, *there exists a constant $C$ depending only on* $(\mu, \sigma, M)$ *such that, for any* $\gamma \geq 0$, *with probability at least* $1 - \exp(-\gamma)$,

$$\|\widehat{R} - R^*\|_p \leq C\left( \sqrt{\frac{\mathrm{Pdim}(\mathcal{F}_{\mathrm{FNN}}^R)\log n}{n}} + \sqrt{\frac{\gamma}{n}} \right).$$

Corollary 2 implies that $\|\widehat{R} - R^*\|_p = O_p\big(\sqrt{\log n/n}\big)$, when the true density-ratio $R^* \in \mathcal{F}_{\mathrm{FNN}}$. This convergence rate is slightly faster than the rate for $\widehat{R}_{\mathrm{KT}}$ given in (18). Moreover, the boundedness assumption for the weights of the neural network functions, as imposed by Kato & Teshima (2021), is not needed. Corollary 2 also shows that, if the target ratio is assumed to belong to the optimization space (or hypothesis space), i.e., $R^* \in \mathcal{F}_{\mathrm{FNN}}^R$ without approximation error, then the convergence rate does not depend on the dimension of the data. In other words, the estimation of $R^*$ does not suffer from the curse of dimensionality. However, this is probably not realistic. Therefore, it is important to consider the approximation error due to the fact that $R^* \notin \mathcal{F}_{\mathrm{FNN}}^R$ in applications.

# 6  Conclusions

In this paper, we have established the non-asymptotic error bounds for the deep density-ratio estimator using the Bregman divergence criterion. Under reasonable conditions, we have shown that the deep density-ratio estimator achieves the optimal minimax convergence rate. When the data is supported on an approximate low-dimensional manifold, we have shown that the neural estimator can mitigate the curse of dimensionality. We have also analyzed the convergence properties of the telescoping density ratio estimator (Rhodes et al., 2020) and provided sufficient conditions under which it has a lower error bound than a single-ratio estimator.

A limitation of this work is that certain boundedness assumptions on the target density ratio such as Assumption 2 or 3 is needed. Also, when the boundedness assumption is partially relaxed as in Assumption 3, the error bound in Theorem 2 is not as sharp as that with the boundedness assumption in Theorem 1. It would be interesting to further relax or remove such assumptions. It would also be useful to improve the error bound in Theorem 2 if possible. These are interesting and challenging problems that deserve further study in the future.

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

# A  Appendix

## A.1  Theoretical Proofs

In the appendix, we provide all the technical details and proofs of the theorems stated in the paper.

**Verification of (1):** Equation (1 holds because

$$
\begin{aligned}
& E_{p^*} \Delta_\psi(R^*(Z), R(Z)) \\
= & \ E_{p^*}[\psi(R^*(Z)) - \psi(R(Z)) - \psi'(R(Z))(R^*(Z) - R(Z))] \\
= & \ E_{p^*}[\psi'(R(Z))R(Z) - \psi(R(Z))] - E_{p^*}[\psi'(R(Z))R^*(Z)] + E_{p^*}[\psi(R^*(Z))] \\
= & \ E_{p^*}[\psi'(R(Z))R(Z) - \psi(R(Z))] - E_{q^*}[\psi'(R(Z))] + E_{p^*}[\psi(R^*(Z))],
\end{aligned}
$$

486    where $E_{p^*}[\psi'(R(Z))R^*(Z)] = E_{q^*}[\psi'(R(Z))]$ by the definition of $R^*$. This verifies (1).

487    We now prove the following lemmas.

488      **Lemma A.1.**     *1. If the convex function $f : \mathbb{R} \to \mathbb{R}$ is $\mu$-smooth over $\mathbb{R}$, then for any*
489        *$x, y \in \mathbb{R}$, the following inequality holds*

$$f(y) \leq f(x) + f'(x)(y - x) + \frac{\mu}{2}(y - x)^2.$$

490       *2. Let $f : \mathbb{R} \to \mathbb{R}$ be a first-order differentiable and convex function. If $f$ is $\sigma$-strongly convex,*
491        *then for any $x, y \in \mathbb{R}$, the following inequality holds*

$$f(y) \geq f(x) + f'(x)(y - x) + \frac{\sigma}{2}(y - x)^2.$$

492    *Proof of Lemma A.1.* The proof of Lemma A.1 is standard and can be found in Beck (2017).      $\square$

493      **Lemma A.2.** *Under Assumptions* 1-2, *we have*

494      *(a). There exist two constants $c_0 = \frac{\sigma e^{-3M}}{2}, C_0 = \frac{\mu e^{3M}}{2}$, such that*

$$c_0 \|D - D^*\|_{\max}^2 \leq \mathcal{B}_\psi\left(e^D\right) - \mathcal{B}_\psi\left(e^{D^*}\right),$$

495       *and*

$$\mathcal{B}_\psi\left(e^D\right) - \mathcal{B}_\psi\left(e^{D^*}\right) \leq C_0 \|D - D^*\|_{\max}^2.$$

496      *(b). For $t_1, t_2 \in [-M, M]$, there exist two constants $C_1, C_2$, such that*

$$|\mathcal{L}_1(t_1) - \mathcal{L}_1(t_2)| \leq C_1|t_1 - t_2|,$$

497       *and*

$$|\mathcal{L}_2(t_1) - \mathcal{L}_2(t_2)| \leq C_2|t_1 - t_2|.$$

498      *Actually, we can take $C_1 = 2e^{2M}\mu, C_2 = e^M\mu$.*

499    *Proof of Lemma A.2.* (a) Let $\Delta_\psi(x, y) := \psi(x) - \psi(y) - \psi'(x)(x - y)$. Since
500    $E_{p^*}\Delta_\psi(e^{D(Z)}, e^{D^*(Z)}) = \mathcal{B}_\psi(e^D) - \mathcal{B}_\psi(e^{D^*})$ and $\psi$ is $\mu$-smooth and $\sigma$-strongly convex, by Lemma
501    A.1,

$$\frac{\sigma}{2}E_{p^*}\{e^{D(Z)} - e^{D^*(Z)}\}^2 \leq E_{p^*}\Delta_\psi(e^{D(Z)}, e^{D^*(Z)}) \leq \frac{\mu}{2}E_{p^*}\{e^{D(Z)} - e^{D^*(Z)}\}^2,$$

502    and then by Assumption 2,

$$\frac{\sigma e^{-2M}}{2}E_{p^*}\{D(Z) - D^*(Z)\}^2 \leq E_{p^*}\Delta_\psi(e^{D(Z)}, e^{D^*(Z)}) \leq \frac{\mu e^{2M}}{2}E_{p^*}\{D(Z) - D^*(Z)\}^2. \text{ (A.1)}$$

503    As $E_{p^*}\{D(Z) - D^*(Z)\}^2 = E_{q^*}e^{-D^*(Z)}\{D(Z) - D^*(Z)\}^2$ and $\|D^*\|_\infty \leq M$, we have

$$e^{-M}E_{q^*}\{D(Z) - D^*(Z)\}^2 \leq E_{p^*}\{D(Z) - D^*(Z)\}^2 \leq e^M E_{q^*}\{D(Z) - D^*(Z)\}^2. \quad \text{(A.2)}$$

504    Let $c_0 = \frac{\sigma e^{-3M}}{2}, C_0 = \frac{\mu e^{3M}}{2}$, then (A.1) and (A.2) imply that

$$c_0 \|D - D^*\|_{\max}^2 \leq \mathcal{B}_\psi\left(e^D\right) - \mathcal{B}_\psi\left(e^{D^*}\right) \leq C_0 \|D - D^*\|_{\max}^2.$$

505    (b) Obviously, for $t_1, t_2 \in [-M, M]$,

$$
\begin{aligned}
|\mathcal{L}_1(t_1) - \mathcal{L}_1(t_2)| &= |\psi'(e^{t_1})e^{t_1} - \psi(e^{t_1}) - (\psi'(e^{t_2})e^{t_2} - \psi(e^{t_2}))| \\
&\leq e^{t_1}|\psi'(e^{t_1}) - \psi'(e^{t_2})| + |\psi(e^{t_1}) - \psi(e^{t_2}) - \psi'(e^{t_2})(e^{t_1} - e^{t_2})| \\
&\leq e^M\mu|e^{t_1} - e^{t_2}| + \frac{\mu}{2}|e^{t_1} - e^{t_2}|^2 \\
&\leq 2e^M\mu|e^{t_1} - e^{t_2}| \quad (As \ |e^{t_1} - e^{t_2}| \leq 2e^M) \\
&\leq 2e^{2M}\mu|t_1 - t_2|.
\end{aligned}
$$

506 and

$$
\begin{aligned}
|\mathcal{L}_2(t_1) - \mathcal{L}_2(t_2)| &= |\psi'(e^{t_1}) - \psi'(e^{t_2})| \\
&\leq \mu|e^{t_1} - e^{t_2}| \\
&\leq e^M \mu|t_1 - t_2|.
\end{aligned}
$$

507 The proof of the lemma is completed. □

508 *Proof of Theorem 1.* For notational convenience, denote $\epsilon_n = \|D_{\mathrm{NN}} - D^*\|_{\max}$ and use $E_I$ to
509 denote $E_{I^*}$, $I = p, q$. Recall that $E_{n_I}$ denotes the expectation with respect to (w.r.t) the empirical
510 distribution of $\{Z_{I,t}\}_{t=1}^{n_I}$ for $I = p, q$. As $\widehat{D} \in \arg\min_{D \in \mathcal{F}_{\mathrm{FNN}}} \mathcal{L}_{n_p, n_q}(D)$, where $\mathcal{L}_{n_p, n_q}(D) =$
511 $1/n_p \sum_{j=1}^{n_p} \mathcal{L}_1(D(Z_{p,j})) + 1/n_q \sum_{i=1}^{n_q} \mathcal{L}_2(D(Z_{q,i}))$, we have

$$
\begin{aligned}
&c_0\|\widehat{D} - D^*\|_{\max}^2 \\
\leq\ & \mathcal{B}_\psi\left(e^{\widehat{D}}\right) - \mathcal{B}_\psi\left(e^{D^*}\right) \\
\leq\ & \mathcal{B}_\psi\left(e^{\widehat{D}}\right) - \mathcal{B}_\psi\left(e^{D^*}\right) - \mathcal{L}_{n_p, n_q}(\widehat{D}) + \mathcal{L}_{n_p, n_q}(D_{\mathrm{NN}}) \\
=\ & \mathcal{B}_\psi\left(e^{\widehat{D}}\right) - \mathcal{L}_{m,n}(\widehat{D}) - \left\{\mathcal{B}_\psi\left(e^{D^*}\right) - \mathcal{L}_{m,n}(D^*)\right\} \\
&+ \left\{\mathcal{L}_{n_p, n_q}(D_{\mathrm{NN}}) - \mathcal{L}_{n_p, n_q}(D^*)\right\} \\
=\ & (E_{p^*} - E_{n_p})\{\mathcal{L}_1(\widehat{D}) - \mathcal{L}_1(D^*)\} + (E_q - E_{n_q})\{\mathcal{L}_2(\widehat{D}) - \mathcal{L}_2(D^*)\} \\
&+ E_{n_p}\{\mathcal{L}_1(D_{\mathrm{NN}}) - \mathcal{L}_1(D^*)\} + E_{n_q}\{\mathcal{L}_2(D_{\mathrm{NN}}) - \mathcal{L}_2(D^*)\}. \qquad (A.3)
\end{aligned}
$$

512 By Theorem 2.1 in Bartlett et al. (2005), with probability at least $1 - \exp(-\gamma_1)$,

$$
E_{n_p}\{\mathcal{L}_1(D_{\mathrm{NN}}) - \mathcal{L}_1(D^*)\} \leq E_p\{\mathcal{L}_1(D_{\mathrm{NN}}) - \mathcal{L}_1(D^*)\} + \sqrt{2}C_1\|D_{\mathrm{NN}} - D^*\|_{\max}\sqrt{\frac{\gamma_1}{n}} + \frac{16C_1 M\gamma_1}{3n}.
$$
$$(A.4)$$

513 Also, with probability at least $1 - \exp(-\gamma_1)$,

$$
E_{n_q}\{\mathcal{L}_2(D_{\mathrm{NN}}) - \mathcal{L}_2(D^*)\} \leq E_q\{\mathcal{L}_2(D_{\mathrm{NN}}) - \mathcal{L}_2(D^*)\} + \sqrt{2}C_2\|D_{\mathrm{NN}} - D^*\|_{\max}\sqrt{\frac{\gamma_1}{n}} + \frac{16C_2 M\gamma_1}{3n}.
$$
$$(A.5)$$

514 The inequalities (A.4) and (A.5) together imply that with probability at least $1 - 2\exp(-\gamma_1)$,

$$
\begin{aligned}
&E_{n_p}\{\mathcal{L}_1(D_{\mathrm{NN}}) - \mathcal{L}_1(D^*)\} + E_{n_q}\{\mathcal{L}_2(D_{\mathrm{NN}}) - \mathcal{L}_2(D^*)\} \\
\leq\ & E_p\{\mathcal{L}_1(D_{\mathrm{NN}}) - \mathcal{L}_1(D^*)\} + E_q\{\mathcal{L}_2(D_{\mathrm{NN}}) - \mathcal{L}_2(D^*)\} \\
&+ \sqrt{2}(C_1 + C_2)\|D_{\mathrm{NN}} - D^*\|_{\max}\sqrt{\frac{\gamma_1}{n}} + \frac{16(C_1 + C_2)M\gamma_1}{3n} \\
=\ & \mathcal{B}_\psi\left(e^{D_{\mathrm{NN}}}\right) - \mathcal{B}_\psi\left(e^{D^*}\right) + \sqrt{\frac{2\gamma_1}{n}}(C_1 + C_2)\|D_{\mathrm{NN}} - D^*\|_{\max} + \frac{16(C_1 + C_2)M\gamma_1}{3n} \\
\leq\ & C_0\|D_{\mathrm{NN}} - D^*\|_{\max}^2 + \sqrt{\frac{2\gamma_1}{n}}(C_1 + C_2)\|D_{\mathrm{NN}} - D^*\|_{\max} + \frac{16(C_1 + C_2)M\gamma_1}{3n}. \quad (A.6)
\end{aligned}
$$

515 **Step 1**. Let $g = (D - D^*)^2$, then $g \leq 4M^2$ by Assumption 2. If $\|D - D^*\|_{\max} \leq r$, then

$$
\begin{aligned}
\mathrm{var}_p(g) \leq E_p(g^2) &= E_p(D - D^*)^4 \\
&\leq 4M^2 E_p(D - D^*)^2 \\
&\leq 4M^2 r^2.
\end{aligned}
$$

516 Regarding $g$ as a function of $D - D^*$, we have

$$
\begin{aligned}
|g(D_1 - D^*) - g(D_2 - D^*)| &= |D_1^2 - 2D_1 D^* - (D_2^2 - 2D_2 D^*)| \\
&= |(D_1 + D_2 - 2D^*)(D_1 - D_2)| \\
&= |(D_1 + D_2 - 2D^*)\{(D_1 - D^*) - (D_2 - D^*)\}| \\
&\leq 4M|(D_1 - D^*) - (D_2 - D^*)|.
\end{aligned}
$$

517 Thus $g$ can be viewed as the function of $D - D^*$ with a Lipschitz constant $4M$. Denote $\mathcal{F}_{\text{FNN}}^{D^*,r} =$
518 $\{D \in \mathcal{F}_{\text{FNN}}, \|D - D^*\|_{\max} \le r\}$, and

$$R_{n_I}\mathcal{F} = \sup_{f \in \mathcal{F}} \frac{1}{n_I} \sum_{i=1}^{n_I} \eta_i^I f(Z_{I,i}), \ I = p, q,$$

519 where $\eta_i^I, i = 1, 2, \ldots, n_I$ are i.i.d. Rademacher variables. For the rest of the proof of Theorem
520 1, we use $E_\eta R_{n_I}\mathcal{F}$ to denote the conditional expectation of $R_{n_I}\mathcal{F}$ w.r.t $\eta_i^I, i = 1, 2, \ldots, n_I$, given
521 $Z_{I,i}, i = 1, 2, \ldots, n_I$ and $E_{I,\eta} R_{n_I}\mathcal{F}$ to denote the expectation of $R_{n_I}\mathcal{F}$ jointly w.r.t $\eta_i^I, Z_{I,i}, i =$
522 $1, 2, \ldots, n_I$. Again, by Theorem 2.1 in Bartlett et al. (2005), with probability at least $1 - \exp(-\gamma_1)$,

$$\|D - D^*\|_{p,n_p}^2 - \|D - D^*\|_p^2$$

$$\le \ 3E_{p,\eta} R_{n_p} \left\{ (D - D^*)^2 : D \in \mathcal{F}_{\text{FNN}}^{D^*,r} \right\} + 2\sqrt{\frac{2\gamma_1}{n}} M + \frac{16M^2}{3} \frac{\gamma_1}{n}$$

$$\le \ 24M E_{p,\eta} R_{n_p} \left\{ (D - D^*) : D \in \mathcal{F}_{\text{FNN}}^{D^*,r} \right\} + 2\sqrt{\frac{2\gamma_1}{n}} Mr + \frac{16M^2}{3} \frac{\gamma_1}{n}, \qquad \text{(A.7)}$$

523 where the last inequality follows from Talagland's contraction theorem. Similarly, with probability at
524 least $1 - \exp(-\gamma_1)$,

$$\|D - D^*\|_{q,n_q}^2 - \|D - D^*\|_q^2 \le 24M E_{q,\eta} R_{n_q} \left\{ (D - D^*) : D \in \mathcal{F}_{\text{FNN}}^{D^*,r} \right\} + 2\sqrt{\frac{2\gamma_1}{n}} Mr + \frac{16M^2}{3} \frac{\gamma_1}{n}. \tag{A.8}$$

525 Let

$$\frac{R_n(r)}{24M} = \max_{I \in \{p,q\}} \left\{ E_{I,\eta} R_{n_I} \left\{ (D - D^*) : D \in \mathcal{F}_{\text{FNN}}^{D^*,r} \right\} \right\}.$$

526 When

$$r^2 \ge R_n(r), r^2 \ge \frac{16M^2\gamma}{3n}, \tag{A.9}$$

527 (A.7) and (A.8) indicate that with probability at least $1 - 2\exp(-\gamma_1)$,

$$\|D - D^*\|_{n_p,n_q}^2 \ = \ \max\{\|D - D^*\|_{p,n_p}^2, \|D - D^*\|_{q,n_q}^2\}$$

$$\le \ \max\{\|D - D^*\|_p^2, \|D - D^*\|_q^2\} + R_n(r) + 2\sqrt{\frac{2\gamma_1}{n}} Mr + \frac{16M^2}{3} \frac{\gamma_1}{n}$$

$$= \ \|D - D^*\|_{\max}^2 + R_n(r) + 2\sqrt{\frac{2\gamma_1}{n}} Mr + \frac{16M^2}{3} \frac{\gamma_1}{n}$$

$$\le \ (2r)^2.$$

528 Thus, when (A.9) holds, with probability at least $1 - 2\exp(-\gamma_1)$,

$$\|D - D^*\|_{\max} \le r \Rightarrow \|D - D^*\|_{n_p,n_q} \le 2r. \tag{A.10}$$

529 **Step** 2. Suppose $\|\widehat{D} - D^*\|_{\max} \le r_0$ and let

$$\mathcal{G}_i = \left\{ \mathcal{L}_i(D) - \mathcal{L}_i(D^*) : D \in \mathcal{F}_{\text{FNN}}^{D^*,r_0} \right\}, i = 1, 2.$$

530 For each $(I, i) \in \{(p, 1), (q, 2)\}$, with probability at least $1 - 2\exp(-\gamma_1)$,

$$(E_I - E_{n_I})\{\mathcal{L}_i(\widehat{D}) - \mathcal{L}_i(D^*)\} \le 6E_\eta R_{n_I} \mathcal{G}_i + \sqrt{2} C_i r_0 \sqrt{\frac{\gamma_1}{n}} + \frac{46 C_i M \gamma_1}{3n}. \tag{A.11}$$

531 Denote $\hat{\mathcal{F}}_{\text{FNN}}^{D^*,r} = \{D \in \mathcal{F}_{\text{FNN}}, \|D - D^*\|_{n_p,n_q} \le r\}$. By (A.10) in *Step* 1, when $r_0^2 \ge R_n(r_0)$ and
532 $r_0^2 \ge 16M^2\gamma_1/(3n)$, with probability at least $1 - 2\exp(-\gamma_1)$, for each $(I, i) \in \{(p, 1), (q, 2)\}$,

$$E_\eta R_{n_I} \mathcal{G}_i \ \le \ 2C_i E_\eta R_{n_I} \left\{ (D - D^*) : D \in \mathcal{F}_{\text{FNN}}^{D^*,r_0} \right\}$$

$$\le \ 2C_i E_\eta R_{n_I} \left\{ (D - D^*) : D \in \hat{\mathcal{F}}_{\text{FNN}}^{D^*,2r_0} \right\}.$$

533   Denote $\hat{\mathcal{F}}_I^{D^*,r} = \{D \in \mathcal{F}_{\text{FNN}}, \|D - D^*\|_{I,n_I} \le r\}$. When $n \ge \text{Pdim}(\mathcal{F}_{\text{FNN}}), r_0 \ge 1/n$ and
534   $n \ge (2eM)^2$, we have

$$E_\eta R_{n_I}\{(D - D^*) : D \in \hat{\mathcal{F}}_I^{D^*,2r_0}\} \le 64r_0\sqrt{\frac{\text{Pdim}(\mathcal{F}_{\text{FNN}})\log n}{n}}, \tag{A.12}$$

535   and thus

$$E_\eta R_{n_I}\mathcal{G}_i \le 128C_i r_0\sqrt{\frac{\text{Pdim}(\mathcal{F}_{\text{FNN}})\log n}{n}}. \tag{A.13}$$

536   Combining (A.3) (A.6) (A.11) and (A.13), with probability at least $1 - 8\exp(-\gamma_1)$, we have

$$
\begin{aligned}
c_0\|\widehat{D} - D^*\|_{\max}^2 \quad \le \quad & 768(C_1 + C_2)r_0\sqrt{\frac{\text{Pdim}(\mathcal{F}_{\text{FNN}})\log n}{n}} \\
& + \sqrt{\frac{2\gamma_1}{n}}(C_1 + C_2)r_0 + \frac{46(C_1 + C_2)M\gamma_1}{3n} + C_0\epsilon_n^2 \\
& + \sqrt{\frac{2\gamma_1}{n}}(C_1 + C_2)\epsilon_n + \frac{16(C_1 + C_2)M\gamma_1}{3n} \\
= \quad & (C_1 + C_2)r_0\left(768\sqrt{\frac{\text{Pdim}(\mathcal{F}_{\text{FNN}})\log n}{n}} + \sqrt{\frac{2\gamma_1}{n}}\right) \\
& + C_0\epsilon_n^2 + \sqrt{\frac{2\gamma_1}{n}}(C_1 + C_2)\epsilon_n + \frac{62(C_1 + C_2)M\gamma_1}{3n}.
\end{aligned}
$$

537   Therefore, when $\max\left\{\sqrt{\text{Pdim}(\mathcal{F}_{\text{FNN}})\log n/n}, \epsilon_n\right\} \ll r_0$, there exists $r_1 \ll r_0$ such that $\|\widehat{D} - $
538   $D^*\|_{\max} \ll r_1$.

539   **Step** 3.   Let $r_* = \inf\{r \ge 0 : R_n(s) \le s^2, \text{for } s \ge r\}$ and $E = $
540   $\left\{\|D - D^*\|_{n_p,n_q} \le 4r_* \text{ for all } D \in \mathcal{F}_{\text{FNN}}^{D^*,2r_*}\right\}$. We intend to prove

$$r_* \le \kappa M\sqrt{\frac{\text{Pdim}(\mathcal{F}_{\text{FNN}})\log n}{n}}, \quad \kappa = 24 \times 130. \tag{A.14}$$

541   When $r_* \le 2\sqrt{3}M\sqrt{\log n/n}/3$, the inequality is trivial. When $r_* \ge 2\sqrt{3}M\sqrt{\log n/n}/3$, by the
542   result in Step 1, $P(E) \ge 1 - 2/n$. As a result,

$$
\begin{aligned}
r_*^2 \quad &\le \quad R_n(r_*) \\
&\le \quad R_n(2r_*) \\
&= \quad 24M \max_{I \in \{p,q\}}\left\{E_{I,\eta}R_{n_I}\{(D - D^*) : D \in \mathcal{F}_{\text{FNN}}^{D^*,2r_*}\}\right\}.
\end{aligned}
$$

543   For each $I \in \{p,q\}$,

$$
\begin{aligned}
E_{I,\eta}R_{n_I}\left\{(D - D^*) : D \in \mathcal{F}_{\text{FNN}}^{D^*,2r_*}\right\} \quad &= \quad E_I E_\eta R_{n_I}\left\{(D - D^*) : D \in \mathcal{F}_{\text{FNN}}^{D^*,2r_*}\right\} \\
&= \quad E_I E_\eta R_{n_I}\left\{(D - D^*) : D \in \mathcal{F}_{\text{FNN}}^{D^*,2r_*}\right\}(I_E + I_{E^c}) \\
&\le \quad E_I E_\eta R_{n_I}\left\{(D - D^*) : D \in \hat{\mathcal{F}}_{\text{FNN}}^{D^*,4r_*}\right\} + \frac{4M}{n}.
\end{aligned}
$$

544   It follows from (A.12) that

$$
\begin{aligned}
r_*^2 \quad &\le \quad 24M\left(128r_*\sqrt{\frac{\text{Pdim}(\mathcal{F}_{\text{FNN}})\log n}{n}} + \frac{4M}{n}\right) \\
&= \quad 24M\left(128r_*\sqrt{\frac{\text{Pdim}(\mathcal{F}_{\text{FNN}})\log n}{n}} + r_* \cdot \frac{4M}{n} \cdot \frac{1}{r_*}\right) \\
&\le \quad 24Mr_*\left(128\sqrt{\frac{\text{Pdim}(\mathcal{F}_{\text{FNN}})\log n}{n}} + \sqrt{\frac{3}{n\log n}}\right) \\
&\le \quad \kappa\sqrt{\frac{\text{Pdim}(\mathcal{F}_{\text{FNN}})\log n}{n}}Mr_*,
\end{aligned}
$$

where $\kappa = 24 \times 130$. Thus, $r_* \leq \kappa M \sqrt{\mathrm{Pdim}(\mathcal{F}_{\mathrm{FNN}}) \log n / n}$ and (A.14) is proved.

***Step* 4.** Let $B_{\max}(D^*, r) = \{D \in \mathcal{F}_{\mathrm{FNN}}, \|D - D^*\|_{\max} \leq r\}$, $\bar{r} \geq \max\left(\sqrt{\log n / n}, r_*\right)$ and $l = \left\lfloor \log_2(2M/\sqrt{\log n / n}) \right\rfloor$. Then, the neural network function space $\mathcal{F}_{\mathrm{FNN}}$ can be divided into

$$B_{\max}(D^*, \bar{r}), B_{\max}(D^*, 2\bar{r}) \backslash B_{\max}(D^*, \bar{r}), \ldots, B_{\max}(D^*, 2^l \bar{r}) \backslash B_{\max}(D^*, 2^{l-1} \bar{r}).$$

As $\bar{r} \geq r_*$, it then follows from the definition of $r_*$ that $\bar{r}^2 \geq R_n(\bar{r})$. Further, if $\bar{r}^2 \geq 16M^2 \gamma_1/(3n)$, according to (A.10) in *Step* 1, with probability at least $1 - 2l \exp(-\gamma_1)$, for any $j = 1, 2, \ldots, l$,

$$\|D - D^*\|_{\max} \leq 2^j \bar{r} \Rightarrow \|D - D^*\|_{n_p, n_q} \leq 2^{j+1} \bar{r}.$$

Suppose that for some $j \leq l$, $\widehat{D} \in B_{\max}(D^*, 2^j \bar{r}) \backslash B_{\max}(D^*, 2^{j-1} \bar{r})$, then by the results in Step 2, with probability at least $1 - 8 \exp(-\gamma_1)$,

$$
\begin{aligned}
c_0 \|\widehat{D} - D^*\|_{\max}^2 &\leq (C_1 + C_2) 2^j \bar{r} \left( 768 \sqrt{\frac{\mathrm{Pdim}(\mathcal{F}_{\mathrm{FNN}}) \log n}{n}} + \sqrt{\frac{2\gamma_1}{n}} \right) \\
&\quad + C_0 \epsilon_n^2 + \sqrt{\frac{2\gamma_1}{n}} (C_1 + C_2) \epsilon_n + \frac{62(C_1 + C_2) M \gamma_1}{3n}.
\end{aligned}
$$

If

$$\frac{1}{c_0}(C_1 + C_2) \left( 768 \sqrt{\frac{\mathrm{Pdim}(\mathcal{F}_{\mathrm{FNN}}) \log N}{N}} + \sqrt{2} \sqrt{\frac{\gamma_1}{N}} \right) \leq \frac{1}{8} 2^j \bar{r}, \tag{A.15}$$

and

$$\frac{1}{c_0} \left[ C_0 \epsilon_N^2 + \sqrt{2}(C_1 + C_2) \epsilon_N \sqrt{\frac{\gamma_1}{N}} + \frac{62(C_1 + C_2) M \gamma_1}{3N} \right] \leq \frac{1}{8} 2^{2j} \bar{r}^2, \tag{A.16}$$

then

$$\|\widehat{D} - D^*\|_{\max}^2 \leq 2^{2j-2} \bar{r}^2 \Leftrightarrow \|\widehat{D} - D^*\|_{\max} \leq 2^{j-1} \bar{r}.$$

In short, to obtain this inequality, we need $\bar{r}$ satisfying (A.15), (A.16) and $\bar{r} \geq \max\left(\sqrt{\log n / n}, 4\sqrt{3} M \sqrt{\gamma_1/n}/3, r_*\right)$. As $r_* \leq \kappa M \sqrt{\mathrm{Pdim}(\mathcal{F}_{\mathrm{FNN}}) \log n / n}$, there exists a constant $C_* = C_*(c_0, C_0, C_1, C_2, M) = C'(\mu, \sigma, M)$ such that

$$\bar{r} = C_* \left( \sqrt{\frac{\mathrm{Pdim}(\mathcal{F}_{\mathrm{FNN}}) \log n}{n}} + \sqrt{\frac{\gamma_1}{n}} + \epsilon_n \right)$$

satisfies all the requirements. As a result, with probability at least $1 - 10l \exp(-\gamma_1)$,

$$\|\widehat{D} - D^*\|_{\max} \leq \bar{r} \text{ and } \|\widehat{D} - D^*\|_{n_p, n_q} \leq 2\bar{r}.$$

Let $\gamma_1 = \log 10l + \gamma, l = \lfloor \log_2(2M/\sqrt{\log n / n}) \rfloor$, there exists $C = C(c_0, C_0, C_1, C_2, M) = C(\mu, \sigma, M)$ such that with probability at least $1 - \exp(-\gamma)$,

$$\|\widehat{D} - D^*\|_{\max} \leq \bar{r} \leq C \left( \sqrt{\frac{\mathrm{Pdim}(\mathcal{F}_{\mathrm{FNN}}) \log n}{n}} + \sqrt{\frac{\gamma}{n}} + \epsilon_n \right),$$

and

$$\|\widehat{D} - D^*\|_{n_p, n_q} \leq 2C \left( \sqrt{\frac{\mathrm{Pdim}(\mathcal{F}_{\mathrm{FNN}}) \log n}{n}} + \sqrt{\frac{\gamma}{n}} + \epsilon_n \right).$$

The proof of Theorem 1 is completed. $\qquad\square$

**Lemma A.3.** *The following excess risk decomposition always holds:*

$$\mathcal{B}_\psi\left(e^{\widehat{D}}\right) - \mathcal{B}_\psi\left(e^{D^*}\right) = \left\{ \mathcal{B}_\psi\left(e^{\widehat{D}}\right) - \inf_{D \in \mathcal{F}_{\mathrm{FNN}}} \mathcal{B}_\psi\left(e^D\right) \right\} + \left\{ \inf_{D \in \mathcal{F}_{\mathrm{FNN}}} \mathcal{B}_\psi\left(e^D\right) - \mathcal{B}_\psi\left(e^{D^*}\right) \right\} \tag{A.17}$$

*Under Assumptions* 1 *and* 3*, when* $n \geq \mathrm{Pdim}(\mathcal{F}_{\mathrm{FNN}})$*, there exist three constants* $C, C_0, C_*$*, with* $C, C_0$ *depending only on* $(\mu, \sigma, M)$ *and* $C_*$ *depending only on* $(\mu, \sigma)$*, such that*

$$E_{p^*,q^*} \left\{ \mathcal{B}_\psi \left( e^{\widehat{D}} \right) - \inf_{D \in \mathcal{F}_{\mathrm{FNN}}} \mathcal{B}_\psi \left( e^D \right) \right\} \leq C \sqrt{\frac{\mathrm{Pdim}(\mathcal{F}_{\mathrm{FNN}}) \log n}{n}}, \tag{A.18}$$

*and*

$$E_{p^*,q^*} \| \widehat{D} - D^* \|_p^2 \leq C_0 \sqrt{\frac{\mathrm{Pdim}(\mathcal{F}_{\mathrm{FNN}}) \log n}{n}} + C_* e^{2M} \inf_{D \in \mathcal{F}_{\mathrm{FNN}}} \| e^D - e^{D^*} \|_p^2.$$

*Proof of Lemma A.3.* To show (A.18) is the key step in the proof of this theorem, thus we focus on the proof of (A.18). Let

$$D_0 \in \operatorname*{arg\,min}_{D \in \mathcal{F}_{\mathrm{FNN}}} \mathcal{B}_\psi \left( e^D \right).$$

Then,

$$
\begin{aligned}
& E_{p^*,q^*} \left\{ \mathcal{B}_\psi \left( e^{\hat{D}} \right) - \inf_{D \in \mathcal{F}_{\mathrm{FNN}}} \mathcal{B}_\psi \left( e^D \right) \right\} \\
= \ & E_{p^*,q^*} \left\{ \mathcal{B}_\psi \left( e^{\hat{D}} \right) - \mathcal{B}_\psi \left( e^{D_0} \right) \right\} \\
\leq \ & E_{p^*,q^*} \left\{ \mathcal{B}_\psi \left( e^{\hat{D}} \right) - \hat{\mathcal{B}}_\psi \left( e^{\hat{D}} \right) + \hat{\mathcal{B}}_\psi \left( e^{\hat{D}} \right) - \hat{\mathcal{B}}_\psi \left( e^{D_0} \right) \right\} \\
& + E_{p^*,q^*} \left\{ \hat{\mathcal{B}}_\psi \left( e^{D_0} \right) - \mathcal{B}_\psi \left( e^{D_0} \right) \right\} \\
\leq \ & 2 E_{p^*,q^*} \left\{ \sup_{D \in \mathcal{F}_{\mathrm{FNN}}} | \hat{\mathcal{B}}_\psi \left( e^D \right) - \mathcal{B}_\psi \left( e^D \right) | \right\}.
\end{aligned} \tag{A.19}
$$

By the symmetrization technique, Talagrand's lemma, (A.12) and the fact that $\|D\|_\infty \leq M$ for any $D \in \mathcal{F}_{\mathrm{FNN}}$, we can easily get the inequality (A.18) through (A.19). $\qquad \square$

*Proof of Theorem 2.* Theorem 2 is a direct corollary of Lemma A.3. We omit the details here. $\qquad \square$

*Proof of Theorem 3.* Since $D^* \in \mathcal{H}^\beta([0,1]^d, M)$ with $\beta = k + a$ where $k \in \mathbb{N}^+$ and $a \in (0,1]$, by Lemma 1, for the $\mathcal{F}_{\mathrm{FNN}}$, a function class consists of ReLU FNN with width $\mathcal{W} = 38(\lfloor \beta \rfloor + 1)^2 d^{\lfloor \beta \rfloor + 1} L \lceil \log_2(8L) \rceil$ and depth $\mathcal{D} = 21(\lfloor \beta \rfloor + 1)^2 K \lceil \log_2(8K) \rceil$, where $K, L \in \mathbb{N}^+$, there exists a function $\phi_0 \in \mathcal{F}_{\mathrm{FNN}}$ such that

$$\sup_{x \in [0,1]^d \setminus H_{B,\delta}} |D^* - \phi_0| \leq 18 M (\lfloor \beta \rfloor + 1)^2 d^{\lfloor \beta \rfloor + (\beta \vee 1)/2} (KL)^{-\frac{2\beta}{d}}, \tag{A.20}$$

where $H_{B,\delta} = \cup_{i=1}^d \{ x = [x_1, \ldots, x_d] : x_i \in \cup_{b=1}^{B-1} (b/B - \delta, b/B) \}, B = \lceil (KL)^{2/d} \rceil, \delta \in (0, 1/(3B)]$. As $D_{\mathrm{NN}} \in \operatorname{arg\,min}_{D \in \mathcal{F}_{\mathrm{FNN}}} \|D - D^*\|_{\max}$, then

$$\|D_{\mathrm{NN}} - D^*\|_{\max}^2 \leq \|\phi_0 - D^*\|_{\max}^2.$$

By the result in (A.20), for $I = p$ or $q$, we have

$$
\begin{aligned}
\|\phi_0 - D^*\|_I^2 & = \int_{[0,1]^d \setminus H_{B,\delta}} |D^* - \phi_0|^2 I^*(x) dx + \int_{H_{B,\delta}} |D^* - \phi_0|^2 I^*(x) dx \\
& \leq 324 M^2 (\lfloor \beta \rfloor + 1)^4 d^{2\lfloor \beta \rfloor + (\beta \vee 1)} (KL)^{-\frac{4\beta}{d}} + 4M^2 \int_{H_{B,\delta}} I^*(x) dx.
\end{aligned}
$$

As $p^*(\cdot), q^*(\cdot)$ are the density functions of some measures on $[0,1]^d$ which are absolutely continuous with respect to the Lebesgue measure and $\delta$ can be arbitrarily small, $\int_{H_{B,\delta}} I_0(x) dx$ is also arbitrarily small. Thus we have

$$\|\phi_0 - D^*\|_I^2 \leq 324 M^2 (\lfloor \beta \rfloor + 1)^4 d^{2\lfloor \beta \rfloor + (\beta \vee 1)} (KL)^{-\frac{4\beta}{d}}$$

584 and

$$
\begin{aligned}
\|D_{\mathrm{NN}} - D^*\|_{\max}^2 \quad &\leq \quad \|\phi_0 - D^*\|_{\max}^2 \\
&\leq \quad 324M^2(\lfloor\beta\rfloor+1)^4 d^{2\lfloor\beta\rfloor+(\beta\vee 1)}(KL)^{-\frac{4\beta}{d}} \\
&= \quad 324M^2 C_1(\beta,d)(KL)^{-\frac{4\beta}{d}}.
\end{aligned}
$$

585 By Corollary 1, there exists a constant $C_1$ only depending on $(\mu,\sigma,M)$ such that

$$
\begin{aligned}
E_{p^*,q^*}\|\widehat{D} - D^*\|_{\max}^2 \quad &\leq \quad C_1\left(\frac{\mathrm{Pdim}(\mathcal{F}_{\mathrm{FNN}})\log n}{n} + \|D_{\mathrm{NN}} - D^*\|_{\max}^2\right) \\
&\leq \quad C_1\left\{\frac{\mathrm{Pdim}(\mathcal{F}_{\mathrm{FNN}})\log n}{n} + 324M^2 C_1(\beta,d)(KL)^{-\frac{4\beta}{d}}\right\} \\
&\leq \quad 324M^2 C_1\left\{\frac{\mathrm{Pdim}(\mathcal{F}_{\mathrm{FNN}})\log n}{n} + C_1(\beta,d)(KL)^{-\frac{4\beta}{d}}\right\}. \text{ (A.21)}
\end{aligned}
$$

586 This completes the proof of the first part of Theorem 3.

587 As for the second part of this theorem, based on Theorem 6 in Bartlett et al. (2019), for a specific
588 ReLU network $f_\phi$, where $\phi$ contains the parameters in the network, there exists a universal constant
589 $C_2$ such that
$$
\mathrm{Pdim}(\mathcal{F}_{\mathrm{FNN}}) \leq C_2 \mathcal{S}\mathcal{D}\log\mathcal{S},
$$

590 where $\mathcal{S}$ is the total number of parameters in the network $f_\phi$. For a ReLU FNN with width
591 $\mathcal{W}$ and depth $\mathcal{D}$, it can be easily checked that $\mathcal{S} = O(\mathcal{W}^2\mathcal{D})$. Now for $\mathcal{W} = 114(\lfloor\beta\rfloor +$
592 $1)^2 d^{\lfloor\beta\rfloor+1}$, $\mathcal{D} = 21(\lfloor\beta\rfloor+1)^2\left\lceil n^{\frac{d}{2(d+2\beta)}}\log_2\left(8n^{\frac{d}{2(d+2\beta)}}\right)\right\rceil$, and $\mathcal{W},\mathcal{D}$ satisfy $O(\mathcal{W}^2\mathcal{D}) =$
593 $O\left((\lfloor\beta\rfloor+1)^6 d^{2\lfloor\beta\rfloor+2}\left\lceil n^{\frac{d}{2(d+2\beta)}}\log^{-3}n\right\rceil\right)$, which means $L=1, K=\left\lceil n^{\frac{d}{2(d+2\beta)}}\right\rceil$, and there exist
594 three universal constants $C_3, C_4, C_5$ such that

$$
\begin{aligned}
&\frac{\mathcal{S}\mathcal{D}\log\mathcal{S}\log n}{n} \\
\leq \quad & C_3\left\{(\lfloor\beta\rfloor+1)^6 d^{2\lfloor\beta\rfloor+2}\left\lceil n^{\frac{d}{2(d+2\beta)}}\log^{-3}n\right\rceil\right\} \\
& \times \left(\log\left[C_3\left\{(\lfloor\beta\rfloor+1)^6 d^{2\lfloor\beta\rfloor+2}\left\lceil n^{\frac{d}{2(d+2\beta)}}\log^{-3}n\right\rceil\right\}\right]\right) \\
& \times \left\{21(\lfloor\beta\rfloor+1)^2\left\lceil n^{\frac{d}{2(d+2\beta)}}\log_2\left(8n^{\frac{d}{2(d+2\beta)}}\right)\right\rceil\log n/n\right\} \\
\leq \quad & \frac{C_4}{n}\left\{(\lfloor\beta\rfloor+1)^8 d^{2\lfloor\beta\rfloor+2}n^{\frac{2d}{2(d+2\beta)}}\log^{-1}n\right\} \\
& \times \left\{6\log(\lfloor\beta\rfloor+1) + 2(\lfloor\beta\rfloor+1)\log d + \frac{d}{2(d+2\beta)}\log n\right\} \\
\leq \quad & \frac{C_4}{n}\left\{(\lfloor\beta\rfloor+1)^8 d^{2\lfloor\beta\rfloor+2}n^{\frac{2d}{2(d+2\beta)}}\log^{-1}n\right\}\left\{6(\lfloor\beta\rfloor+1) + 2(\lfloor\beta\rfloor+1)d + \log n\right\} \\
\leq \quad & C_5(\lfloor\beta\rfloor+1)^9 d^{2\lfloor\beta\rfloor+3}n^{-\frac{2\beta}{d+2\beta}}. \qquad\qquad\qquad\qquad\qquad\qquad\qquad\text{(A.22)}
\end{aligned}
$$

595 It follows from (A.21) that

$$
\begin{aligned}
& E_{p^*,q^*}\|\widehat{D} - D^*\|_{\max}^2 \\
\leq \quad & 324M^2 C_1\left\{\frac{\mathrm{Pdim}(\mathcal{F}_{\mathrm{FNN}})\log n}{n} + C_1(\beta,d)(KL)^{-\frac{4\beta}{d}}\right\} \\
\leq \quad & 324M^2 C_1\left\{\frac{C_2\mathcal{S}\mathcal{D}\log\mathcal{S}\log n}{n} + C_1(\beta,d)(KL)^{-\frac{4\beta}{d}}\right\} \\
\leq \quad & 324M^2 C_1\left\{C_2 C_5(\lfloor\beta\rfloor+1)^9 d^{2\lfloor\beta\rfloor+3}n^{-\frac{2\beta}{d+2\beta}} + (\lfloor\beta\rfloor+1)^4 d^{2\lfloor\beta\rfloor+(\beta\vee 1)}n^{-\frac{2\beta}{d+2\beta}}\right\} \\
\leq \quad & 324M^2 C_1(C_2 C_5+1)(\lfloor\beta\rfloor+1)^9 d^{2\lfloor\beta\rfloor+(\beta\vee 3)}n^{-\frac{2\beta}{d+2\beta}}.
\end{aligned}
$$

596 This completes the proof of the second part of Theorem 3. $\qquad\qquad\qquad\qquad\square$

*Proof of Theorem 4.* Based on Theorem 3.1 in Baraniuk & Wakin (2009), there exists a linear projection $A \in \mathbb{R}^{d_\delta \times d}$ such that $AA^T = dI_{d_\delta}/d_\delta$, where $I_{d_\delta} \in \mathbb{R}^{d_\delta \times d_\delta}$ is an identity matrix, and for any $x, y \in \mathcal{M}$,

$$(1-\delta)\|x-y\|_2 \leq \|Ax - Ay\|_2 \leq (1+\delta)\|x-y\|_2. \tag{A.23}$$

Then we have

$$A(\mathcal{M}_\rho) \subseteq A\left([0,1]^d\right) \subseteq \left[-\frac{d}{\sqrt{d_\delta}}, \frac{d}{\sqrt{d_\delta}}\right]^{d_\delta}.$$

Note that for any $z \in A(\mathcal{M})$, there exits a unique $x \in \mathcal{M}$ such that $z = Ax$. Otherwise, suppose we can find $x, x' \in \mathcal{M}, x \neq x'$ such that $z = Ax = Ax'$, then by (A.23), we know $\|x - x'\|_2 = 0$ and thus $x = x'$, which contradicts the assumption that $x \neq x'$. This uniqueness allows us to define a linear operator $\mathcal{SL} : A(\mathcal{M}) \to \mathcal{M}$ such that $A[\mathcal{SL}(z)] = z$. By (A.23), we have

$$(1-\delta)\|\mathcal{SL}(z_1) - \mathcal{SL}(z_2)\|_2 \leq \|z_1 - z_2\|_2 \leq (1+\delta)\|\mathcal{SL}(z_1) - \mathcal{SL}(z_2)\|_2.$$

This implies that the norm of $\mathcal{SL}$ belongs to $[1/(1+\delta), 1/(1-\delta)]$. For the high-dimensional function $D^* : [0,1]^d \to \mathbb{R}$ whose support is $\mathcal{M}_\rho$, it has a approximate low-dimensional representation $\tilde{D}^*$ as follows:

$$\tilde{D}^*(z) = D^*(\mathcal{SL}(z)), \ \forall \, z \in A(\mathcal{M}).$$

As $D^* \in \mathcal{H}^\beta([0,1]^d, M)$ with $\beta = k + a$ where $k \in \mathbb{N}^+$ and $a \in (0,1]$, we have $\tilde{D}^* \in \mathcal{H}^\beta\left(A(\mathcal{M}), M/(1-\delta)^\beta\right)$. By the extended version of Whitney's extension theorem in Fefferman (2006), since $A(\mathcal{M}) \subseteq A\left([0,1]^d\right) \subseteq \left[-d/\sqrt{d_\delta}, d/\sqrt{d_\delta}\right]^{d_\delta}$, there exists $\tilde{D}^*_E \in \mathcal{H}^\beta\left(\left[-d/\sqrt{d_\delta}, d/\sqrt{d_\delta}\right]^{d_\delta}, M/(1-\delta)^\beta\right)$ such that $\tilde{D}^*_E \equiv \tilde{D}^*$ on $A(\mathcal{M})$. If $\mathcal{W} = 38(\lfloor\beta\rfloor + 1)^2 d_\delta^{\lfloor\beta\rfloor + 1} L\lceil\log_2(8L)\rceil$ and $\mathcal{D} = 21(\lfloor\beta\rfloor + 1)^2 K\lceil\log_2(8K)\rceil$, by the first result of Lemma 1, there exists a function $\phi_0$ implemented by a ReLU network with width $\mathcal{W}$ and depth $\mathcal{D}$ such that

$$\sup_{z \in [0,1]^{d_\delta} \setminus H^{d_\delta}_{B,\epsilon}} \left| \tilde{D}^*_E\left(\frac{2dz - d\mathbf{1}_{d_\delta}}{\sqrt{d_\delta}}\right) - \phi_0(z) \right| \leq \frac{18M}{(1-\delta)^\beta}(\lfloor\beta\rfloor + 1)^2 (2d)^\beta d_\delta^{\lfloor\beta\rfloor + (\beta\vee1+\beta)/2}(KL)^{-\frac{2\beta}{d_\delta}}.$$

where $H^{d_\delta}_{B,\epsilon} = \cup_{i=1}^{d_\delta}\left\{x = [x_1, x_2, \ldots, x_{d_\delta}] : x_i \in \cup_{b=1}^{B-1} (b/B - \epsilon, b/B)\right\}$ and $B = \lceil(KL)^{2/d}\rceil, \epsilon \in (0, 1/(3B)]$. Thus

$$\sup_{z \in \left[-\frac{d}{\sqrt{d_\delta}}, \frac{d}{\sqrt{d_\delta}}\right]^{d_\delta} \setminus \tilde{H}^{d_\delta}_{B,\epsilon}} \left| \tilde{D}^*_E(z) - \phi_0\left(\frac{\sqrt{d_\delta}z + d\mathbf{1}_{d_\delta}}{2d}\right) \right|$$
$$\leq \frac{18M}{(1-\delta)^\beta}(\lfloor\beta\rfloor + 1)^2 (2d)^\beta d_\delta^{\lfloor\beta\rfloor + (\beta\vee1+\beta)/2}(KL)^{-\frac{2\beta}{d_\delta}},$$

where $\tilde{H}^{d_\delta}_{B,\epsilon} = \left\{(2dt - d\mathbf{1}_{d_\delta})/\sqrt{d_\delta} : t \in H^{d_\delta}_{B,\epsilon}\right\}$.

Let $\tilde{\phi}_0(x) = \phi_0\left((\sqrt{d_\delta}Ax + d\mathbf{1}_{d_\delta})/(2d)\right)$ and $\bar{H}^d_{*B,\epsilon} = \left\{x \in [0,1]^{d \times d} : (\sqrt{d_\delta}Ax + d\mathbf{1}_{d_\delta})/(2d) \in H^{d_\delta}_{B,\epsilon}\right\}$. It can be easily checked that $\tilde{\phi}_0$ is also a function implemented by a ReLU network with the same structure as $\phi_0$, except that the input layer of $\tilde{\phi}_0$ has $d$ units, instead of $d_\delta$ units. For any $x \in \mathcal{M}_\rho \setminus \bar{H}^d_{*B,\epsilon}$, $Ax \in \left[-d/\sqrt{d_\delta}, d/\sqrt{d_\delta}\right]^{d_\delta} \setminus \tilde{H}^{d_\delta}_{B,\epsilon}$ and there exists a $x' \in \mathcal{M}$ satisfying $\|x - x'\|_2 \leq \rho$. Since $\tilde{D}^*_E \in \mathcal{H}^\beta\left(\left[-d/\sqrt{d_\delta}, d/\sqrt{d_\delta}\right]^{d_\delta}, M/(1-\delta)^\beta\right)$

622 and $D^* \in \mathcal{H}^\beta([0,1]^d, M)$,

$$
\begin{aligned}
&|\tilde{\phi}_0(x) - D^*(x)| \\
&\leq \quad |\tilde{\phi}_0(x) - \tilde{D}_E^*(Ax)| + |\tilde{D}_E^*(Ax) - \tilde{D}_E^*(Ax')| + |\tilde{D}_E^*(Ax') - D^*(x)| \\
&\leq \quad \frac{18M}{(1-\delta)^\beta}(\lfloor\beta\rfloor+1)^2(2d)^\beta d_\delta^{\lfloor\beta\rfloor+(\beta\vee 1+\beta)/2}(KL)^{-\frac{2\beta}{d_\delta}} + \frac{M}{(1-\delta)^\beta}\|Ax'-Ax\|_2 + \rho M \\
&\leq \quad \frac{18M}{(1-\delta)^\beta}(\lfloor\beta\rfloor+1)^2(2d)^\beta d_\delta^{\lfloor\beta\rfloor+(\beta\vee 1+\beta)/2}(KL)^{-\frac{2\beta}{d_\delta}} + \frac{M\sqrt{d}}{(1-\delta)^\beta\sqrt{d_\delta}}\rho + \rho M \\
&\leq \quad \frac{18M}{(1-\delta)^\beta}(\lfloor\beta\rfloor+1)^2(2d)^\beta d_\delta^{\lfloor\beta\rfloor+(\beta\vee 1+\beta)/2}(KL)^{-\frac{2\beta}{d_\delta}} + \frac{2M\sqrt{d}}{(1-\delta)^\beta\sqrt{d_\delta}}\rho \\
&\leq \quad \frac{20M}{(1-\delta)^\beta}(\lfloor\beta\rfloor+1)^2(2d)^\beta d_\delta^{\lfloor\beta\rfloor+(\beta\vee 1+\beta)/2}(KL)^{-\frac{2\beta}{d_\delta}}, \quad\quad\quad\quad\quad\quad\quad\quad (A.24)
\end{aligned}
$$

623 where the last inequality holds when $\rho \leq (\lfloor\beta\rfloor+1)^2 2^\beta d^{\beta-\frac{1}{2}} d_\delta^{\lfloor\beta\rfloor+(\beta-1/2)\vee(1/2)}(KL)^{-\frac{2\beta}{d_\delta}}$. As
624 $D_{\text{NN}} \in \arg\min_{D\in\mathcal{F}_{\text{FNN}}}\|D-D^*\|_{\max}$,

$$
\|D_{\text{NN}}-D^*\|_{\max}^2 \leq \|\tilde{\phi}_0-D^*\|_{\max}^2.
$$

625 By the result in (A.24), for $I = p$ or $q$, it holds

$$
\begin{aligned}
\|\tilde{\phi}_0-D^*\|_I^2 &= \int_{[0,1]^d\backslash H_{B,\delta}}|D^*-\tilde{\phi}_0|^2 I^*(x)dx + \int_{H_{B,\delta}}|D^*-\tilde{\phi}_0|^2 I^*(x)dx \\
&\leq \frac{400M^2}{(1-\delta)^{2\beta}}(\lfloor\beta\rfloor+1)^4(2d)^{2\beta}d_\delta^{\beta\vee 1+3\beta}(KL)^{-\frac{4\beta}{d_\delta}} + \frac{4M^2}{(1-\delta)^{2\beta}}\int_{\bar{H}_{*B,\epsilon}^d} I^*(x)dx.
\end{aligned}
$$

626 As $p^*(\cdot), q^*(\cdot)$ are the density functions of some measures on $[0,1]^d$ which are absolutely continuous
627 w.r.t the Lebesgue measure and $\epsilon$ can be arbitrarily small for the given $\delta$, $\int_{\bar{H}_{*B,\epsilon}^d} I_0(x)dx$ is also
628 arbitrarily small for the given $\delta$. Thus we have

$$
\|\tilde{\phi}_0-D^*\|_I^2 \leq \frac{400M^2}{(1-\delta)^{2\beta}}(\lfloor\beta\rfloor+1)^4(2d)^{2\beta}d_\delta^{\beta\vee 1+3\beta}(KL)^{-\frac{4\beta}{d_\delta}}
$$

629 and

$$
\begin{aligned}
\|D_{\text{NN}}-D^*\|_{\max}^2 &\leq \|\phi_0-D^*\|_{\max}^2 \\
&\leq \frac{400M^2}{(1-\delta)^{2\beta}}(\lfloor\beta\rfloor+1)^4(2d)^{2\beta}d_\delta^{\beta\vee 1+3\beta}(KL)^{-\frac{4\beta}{d_\delta}} \\
&= \frac{400M^2}{(1-\delta)^{2\beta}}C_2(\beta,d,d_\delta)(KL)^{-\frac{4\beta}{d_\delta}}.
\end{aligned}
$$

630 By Corollary 1, there exists a constant $C_1$ only depending on $(\mu, \sigma, M)$, such that

$$
\begin{aligned}
&E_{p^*,q^*}\|\widehat{D}-D^*\|_{\max}^2 \\
&\leq \quad C_1\left(\frac{\text{Pdim}(\mathcal{F}_{\text{FNN}})\log n}{n} + \|D_{\text{NN}}-D^*\|_{\max}^2\right) \\
&\leq \quad C_1\left\{\frac{\text{Pdim}(\mathcal{F}_{\text{FNN}})\log n}{n} + \frac{400M^2}{(1-\delta)^{2\beta}}C_2(\beta,d,d_\delta)(KL)^{-\frac{4\beta}{d_\delta}}\right\} \\
&\leq \quad \frac{400M^2 C_1}{(1-\delta)^{2\beta}}\left\{\frac{\text{Pdim}(\mathcal{F}_{\text{FNN}})\log n}{n} + C_2(\beta,d,d_\delta)(KL)^{-\frac{4\beta}{d_\delta}}\right\}. \quad\quad (A.25)
\end{aligned}
$$

631 For

$$
\mathcal{W} = 114(\lfloor\beta\rfloor+1)^2 d_\delta^{\lfloor\beta\rfloor+1},
$$

632

$$
\mathcal{D} = 21(\lfloor\beta\rfloor+1)^2\left\lceil n^{\frac{d_\delta}{2(d_\delta+2\beta)}}\log_2\left(8n^{\frac{d_\delta}{2(d_\delta+2\beta)}}\right)\right\rceil,
$$

633 and $\mathcal{W}, \mathcal{D}$ satisfy

$$\mathcal{O}(\mathcal{W}^2\mathcal{D}) = \mathcal{O}\left(\left(\lfloor\beta\rfloor + 1\right)^6 d_\delta^{2\lfloor\beta\rfloor+2}\left\lceil n^{\frac{d_\delta}{2(d_\delta+2\beta)}}\log^{-3}n\right\rceil\right),$$

634 along the derivation of (A.22), there exists a universal constants $C^*$ such that

$$\frac{\mathrm{Pdim}(\mathcal{F}_{\mathrm{FNN}})\log n}{n} \leq C^*(\lfloor\beta\rfloor + 1)^9 d_\delta^{2\lfloor\beta\rfloor+3} n^{-\frac{2\beta}{d_\delta+2\beta}}.$$

635 Based on the result of (A.25),

$$
\begin{aligned}
&E_{p^*,q^*}\|\widehat{D} - D^*\|_{\max}^2 \\
&\leq \frac{400M^2C_1}{(1-\delta)^{2\beta}}\left\{\frac{\mathrm{Pdim}(\mathcal{F}_{\mathrm{FNN}})\log n}{n} + C_2(\beta, d, d_\delta)(KL)^{-\frac{4\beta}{d_\delta}}\right\} \\
&\leq \frac{400M^2C_1}{(1-\delta)^{2\beta}}\left\{C^*(\lfloor\beta\rfloor + 1)^9 d_\delta^{2\lfloor\beta\rfloor+3} n^{-\frac{2\beta}{d_\delta+2\beta}} + C_2(\beta, d, d_\delta)n^{-\frac{2\beta}{d_\delta+2\beta}}\right\} \\
&\leq \frac{800M^2C_1C^*}{(1-\delta)^{2\beta}}(\lfloor\beta\rfloor + 1)^9\max\left\{d_\delta^{2\lfloor\beta\rfloor+3}, (2d)^{2\beta}d_\delta^{\beta\vee 1+3\beta}\right\}n^{-\frac{2\beta}{d_\delta+2\beta}} \\
&= \frac{800M^2C_1C^*C_3(\beta, d, d_\delta)}{(1-\delta)^{2\beta}}n^{-\frac{2\beta}{d_\delta+2\beta}}.
\end{aligned}
$$

636 This completes the proof of the theorem and (12). □

637 *Proof of Proposition 1.* For $k = 0, \ldots, K - 2$, the densities $q_k(), q_{k+1}()$ of the synthetic data
638 $\{Z_{k,j}\}_{j=1}^n$ and $\{Z_{k+1,j}\}_{j=1}^n$ satisfy

$$\frac{q_k(t)}{q_{k+1}(t)} = \frac{(1-\alpha_k)q^*(z) + \alpha_k p^*(z)}{(1-\alpha_{k+1})q^*(z) + \alpha_{k+1}p^*(z)} \in \left[\frac{(1-e^{-M})\alpha_k + e^{-M}}{(1-e^{-M})\alpha_{k+1} + e^{-M}}, \frac{1-\alpha_k}{1-\alpha_{k+1}}\right].$$

639 As $\|f\|_2 = \left(\int_{\mathcal{Z}} f^2(x)dx\right)^{\frac{1}{2}}$, then for any density $g$ satisfying $g \geq c$, $\|f\|_2 = \left(\int_{\mathcal{Z}} f^2(x)dx\right)^{\frac{1}{2}} \leq$
640 $\left(\int_{\mathcal{Z}} f^2(x)g(x)/cdx\right)^{\frac{1}{2}} = \|f\|_g/\sqrt{c}$. Using an appropriate $\mathcal{F}_{\mathrm{FNN}}^0$ whose element $D$ satisfies $\|D\|_\infty \leq$
641 $M_0$, for the direct estimate $\widehat{D}_{\mathrm{SRE}}$, as $\log(q^*/p^*)$ is only bounded from below by $-M_0$, by Theorem
642 2, we have

$$\limsup_{n\to\infty} E_{p^*,q^*}\|\widehat{D}_{\mathrm{SRE}} - D^*\|_2 \leq e^{M_0}C_*(\mu, \sigma, c_1)\|R^* - R_{M_0}^*\|_p.$$

643 For $k = 0, 1, \ldots, K - 2$, as $|\log\{q_k(t)/q_{k+1}(t)\}|$ is bounded by $M_0$, by Corollary 1, we have

$$\limsup_{n\to\infty} E_{p^*,q^*}\|\widehat{D}_k - D_k^*\|_2 = 0.$$

644 Let $R_{K-1,M_0}^* = (1-\alpha_{K-1})R_{M_0}^* + \alpha_{K-1}$. As the logarithm of $R_{K-1}^* = (1-\alpha_{K-1})q^*/p^* + \alpha_{K-1}$
645 is also only bounded from below by $-M_0$, again, by Theorem 2,

$$
\begin{aligned}
\limsup_{n\to\infty} E_{p^*,q^*}\|\widehat{D}_{K-1} - D_{K-1}^*\|_2 &\leq e^{M_0}C_*(\mu, \sigma, c_1)\|R_{K-1}^* - R_{K-1,M_0}^*\|_p \\
&= (1-\alpha_{K-1})e^{M_0}C_*(\mu, \sigma, c_1)\|R^* - R_{M_0}^*\|_p.
\end{aligned}
$$

646 Thus

$$
\begin{aligned}
\limsup_{n\to\infty} E_{p^*,q^*}\|\widehat{D}_{\mathrm{TRE}} - D^*\|_2 &\leq \sum_{k=0}^{K-1}\limsup_{n\to\infty} E_{p^*,q^*}\|\widehat{D}_k - D_k^*\|_2 \\
&= \limsup_{n\to\infty} E_{p^*,q^*}\|\widehat{D}_{K-1} - D_{K-1}^*\|_2 \\
&\leq (1-\alpha_{K-1})e^{M_0}C_*(\mu, \sigma, c_1)\|R^* - R_{M_0}^*\|_p.
\end{aligned}
$$

647 This completes the proof of Proposition 1. □

## A.2 Examples of Hölder function class

Let $p^*$ be the density function of a truncated $d$-dimensional multivariate Gaussian with mean zero and covariance matrix $\Sigma_p \in \mathbb{R}^{d \times d}$ in $[0,1]^d$. That means

$$p^*(z) = \exp(-z'\Sigma_p^{-1}z/2)/c(\Sigma_p), \ \ c(\Sigma_p) = \int_{[0,1]^d} \exp(-t'\Sigma_p^{-1}t/2)dt, \ z \in [0,1]^d.$$

Similarly, let

$$q^*(z) = \exp(-z'\Sigma_q^{-1}z/2)/c(\Sigma_q)$$

for some positive definite matrix $\Sigma_q$. For any matrix $A \in \mathbb{R}^{d \times d}$, $A_{i,\cdot}$ is the $i$th row of $A$ for $i = 1, 2, \ldots, d$ and

$$\|A\|_{2,\infty} := \sup_{\|z\|_\infty \le 1} \|Az\|_2.$$

Then,

$$D^*(z) = \log \frac{q^*(z)}{p^*(z)} = \frac{1}{2}z'(\Sigma_p^{-1} - \Sigma_q^{-1})z + \log(c(\Sigma_p) - c(\Sigma_q)), \ z \in [0,1]^d.$$

Let $M = \max \left\{ \frac{1}{2}(\|\Sigma_p^{-1/2}\|_{2,\infty}^2 + \|\Sigma_q^{-1/2}\|_{2,\infty}^2) + |\log[c(\Sigma_p) - c(\Sigma_q)]|, \|(\Sigma_p^{-1} - \Sigma_q^{-1})_{i,\cdot}\|_2, i = 1, 2, \ldots, d \right\}$.
It is straightforward to check that

$$D^* \in \mathcal{H}^2([0,1]^d, M).$$

It implies the Hölder smoothness parameter $\beta$ is 2 for this example.

Moreover, the truncated multivariate Gaussian distributions considered above are special cases of the exponential distribution class defined below. Define the density function class

$$\mathrm{Exp}(\beta, B) := \left\{ p(z) = \exp(g(z))/c_g : z \in [0,1]^d, c_g = \int_{[0,1]^d} \exp(g(t))dt, g \in \mathcal{H}^\beta([0,1]^d, B) \right\}.$$

Suppose that $\Sigma \in \mathbb{R}^{d \times d}$ is positive definite and let $g(z) = z'\Sigma z/2$. Then, $g \in \mathcal{H}^2([0,1]^d, M_\Sigma)$, where $M_\Sigma = \max \left\{ \frac{1}{2}(\|\Sigma^{1/2}\|_{2,\infty}^2, \|\Sigma_{i,\cdot}\|_2, i = 1, 2, \ldots, d \right\}$. If $p^*, q^* \in \mathrm{Exp}(\beta, B)$, we have $D^*(z) = \log[q^*(z)/p^*(z)] \in \mathcal{H}^\beta([0,1]^d, 4B)$.

## A.3 Extension to unbounded support case

In fact, our Theorem 1, Corollary 1 and Theorem 2 do not rely on the hypercube assumption. To relax the hypercube assumption to allow unbounded support, we need to study the upper bound for the approximation error $\|D_{\mathrm{NN}} - D^*\|_{\max}$ carefully. With unbounded support, we may bound $\|D_{\mathrm{NN}} - D^*\|_{\max}$ by the truncation technique under some additional assumptions, at a small price of an additional logarithm term in the error bound.

Specifically, when the pdfs are supported on $\mathbb{R}^d$, to bound the approximation error as in Theorem 3, aside from Assumptions 1-2 and the Hölder class assumption, we need to further assume that $\max\{E_{p^*}I(\|Z\|_\infty \ge \log n), E_{q^*}I(\|Z\|_\infty \ge \log n)\} \le n^{-\frac{2\beta}{d+2\beta}}$. For $I = p$ or $q$, and any $D \in \mathcal{F}_{\mathrm{FNN}}$, where $\mathcal{F}_{\mathrm{FNN}}$ is the function class of ReLU FNNs with width $\mathcal{W}$ and depth $\mathcal{D}$ specified by

$$\mathcal{W} = 114(\lfloor\beta\rfloor + 1)^2 d^{\lfloor\beta\rfloor+1}, \ \ \mathcal{D} = 21(\lfloor\beta\rfloor + 1)^2 \left\lceil n^{\frac{d}{2(d+2\beta)}} \log_2\left(8n^{\frac{d}{2(d+2\beta)}}\right)\right\rceil,$$

we have

$$E_{I^*}[D(Z) - D^*(Z)]^2$$
$$\le E_{I^*}[\{D(Z) - D^*(Z)\}^2 I(\|Z\|_\infty \ge \log n)] + E_{I^*}[\{D(Z) - D^*(Z)\}^2 I(\|Z\|_\infty \le \log n)]$$
$$\le 4M^2 E_{I^*}I(\|Z\|_\infty \ge \log n) + E_{I^*}[\{D(Z) - D^*(Z)\}^2 I(\|Z\|_\infty \le \log n)],$$

where the second inequality follows from the facts that $\|D^*\|_\infty \le M, \|D\|_\infty \le M$ under Assumption 2. Since $D^* \in \mathcal{H}^\beta(\mathbb{R}^d, M)$, $D^*(2t\log n - \log n\mathbf{1}_d) \in \mathcal{H}^\beta([0,1]^d, (2\log n)^{\lfloor\beta\rfloor}M)$ as a function of $t$, where $\mathbf{1}_d$ is the $d$-dimensional all-one vector. By Lemma 1, there exists a function $\phi_0 \in \mathcal{F}_{\mathrm{FNN}}$ such that

$$\sup_{t \in [0,1]^d \setminus H_{B,\delta}} |D^*(2t\log n - \log n\mathbf{1}_d) - \phi_0| \le 18(2\log n)^{\lfloor\beta\rfloor}M(\lfloor\beta\rfloor + 1)^2 d^{\lfloor\beta\rfloor+(\beta\vee 1)/2}n^{-\frac{\beta}{d+2\beta}},$$

 where $H_{B,\delta} = \cup_{i=1}^{d}\big\{t = [t_1, \ldots, t_d] : t_i \in \cup_{b=1}^{B-1}(b/B - \delta, b/B)\big\}, B = \lceil n^{\frac{1}{d+2\beta}}\rceil, \delta \in$
(0, 1/(3B)]. Thus

$$\sup_{z\in[-\log n,\log n]^d\setminus\tilde{H}_{B,\epsilon}^d} \left| D^*(z) - \phi_0\left(\frac{z + \log n\mathbf{1}_d}{2\log n}\right)\right| \le 18(2\log n)^{\lfloor\beta\rfloor}M(\lfloor\beta\rfloor+1)^2 d^{\lfloor\beta\rfloor+(\beta\vee1)/2}n^{-\frac{\beta}{d+2\beta}},$$

where $\tilde{H}_{B,\delta}^d = \left\{2t\log n - \log n : t \in H_{B,\delta}^d\right\}$. Let $\tilde{\phi}_0(z) = \phi_0\left(\frac{z+\log n\mathbf{1}_d}{2\log n}\right) \in \mathcal{F}_{\mathrm{FNN}}$. As $\delta$ can be
arbitrarily small, it then follows from similar lines as in the proof of Theorem 3 that

$$E_{I^*}[\{\tilde{\phi}_0(Z) - D^*(Z)\}^2 I(\|Z\|_\infty \le \log n)] \le 324M^2(\lfloor\beta\rfloor+1)^4 d^{2\lfloor\beta\rfloor+(\beta\vee1)}(2\log n)^{2\lfloor\beta\rfloor}n^{-\frac{2\beta}{d+2\beta}}.$$

Since $D_{\mathrm{NN}} \in \arg\min_{D\in\mathcal{F}_{\mathrm{FNN}}}\|D - D^*\|_{\max}$, we have

$$
\begin{aligned}
\|D_{\mathrm{NN}} - D^*\|_{\max}^2 &\le \|\tilde{\phi}_0 - D^*\|_{\max}^2 \\
&\le \max_{I=p,q}\{4M^2 E_{I^*}I(\|Z\|_\infty \ge \log n) + E_{I^*}[\{\tilde{\phi}_0(Z) - D^*(Z)\}^2 I(\|Z\|_\infty \le \log n)]\} \\
&\le 328M^2(\lfloor\beta\rfloor+1)^4 d^{2\lfloor\beta\rfloor+(\beta\vee1)}(2\log n)^{2\lfloor\beta\rfloor}n^{-\frac{2\beta}{d+2\beta}}.
\end{aligned}
$$

Compared with the upper bound of the approximation error in Theorem 3, when the pdfs are supported
on $\mathbb{R}^d$ (unbounded case), a similar approximation error upper bound can be derived with an additional
logrithmic factor $(2\log n)^{2\lfloor\beta\rfloor}$.

## A.4    Simulation setting and implementation details

Our simulation settings are as follows.

- Beta setting: Let $Z = (Z_1, Z_2, \ldots, Z_p)^\top \in \mathbb{R}^p$ be a random vector of interest, where
  $Z_1, Z_2, \ldots, Z_p$ are i.i.d. random variables following Beta distribution, denoted by
  $\mathrm{Beta}(\alpha, \beta)$. Set $p^*$ as the p.d.f of $\mathrm{Beta}(2.2, 1.5)$ and $q^*$ as the p.d.f of $\mathrm{Beta}(2, 2)$. In
  this setting, we set $p = 5$.

- Normal setting: Let $Z = (Z_1, Z_2, \ldots, Z_d, Z_{d+1}, Z_{d+2}, \ldots, Z_{2d})^\top \in \mathbb{R}^{2d}$ be some random
  vector of interest. Let $p^*$ be the p.d.f of $N(0, I_{2d})$ and $q^*$ be the p.d.f of $N(0, \Sigma(\rho))$, where
  $\Sigma(\rho) = (\sigma_{i,j}^\rho) \in \mathbb{R}^{2d\times 2d}$ and

$$\sigma_{i,j}^\rho = \begin{cases} 1, & i = j; \\ \rho, & |i-j| = d, i,j = 1,2,\ldots,2d; \\ 0, & \text{otherwise.} \end{cases}$$

    In this setting, we set $d = 5$ and $\rho = 0.9$.

We apply the Adam algorithm (Kingma & Ba, 2014) in Pytorch with a learning rate $lr = 0.0001$ and
a weight decay parameter $wd = 0.0001$. A neural network with 2 hidden layers with widths $(64, 64)$
and ReLU activation function, is used in the experiment. The maximum number of epoches is 20000.
In this experiment, the training data size $n$ is 5000 (10000). A validation data is used. The batch size
is 500 (1000), and an early-stopping technique is applied with *patience* = 100 for Beta setting and
*patience* = 1000 for Normal setting, where *patience* is the number of epochs until termination if no
improvement is made on the validation dataset. The experiment is conducted on a laptop with an
*Intel(R) Core(TM) i7-8750H @ 2.20GHz* CPU having 6 cores. We use the LR-Bregman divergence in
this example. For the sequence $0 = \alpha_0 < \alpha_1 < \cdots < \alpha_{K-1} < \alpha_K = 1$, we use the linearly spaced
$\alpha_k$'s, that is $\alpha_k = k/K$, $k = 0, 1, 2, \ldots, K$.

## A.5    The MNIST dataset

We now apply the proposed mixing chain (13) for density ratio estimation to the MNIST dataset
(LeCun et al., 2010). In the implementation, to accelerate the computation, we use the subsampling
method with a training subsample size of 20,000 and a relatively small DenseNet network structure
(Huang et al., 2017); see Table A.1 for the specification of the network architectures. Similarly
to the results in Table 1 of Rhodes et al. (2020), we calculate the average negative log-likelihood
(ANLL) in bits per dimension (bpd, smaller is better). We denote the estimate based on the proposed

mixing chain (13) with the chain length $B$ by "mTRE-$B$". The batch size is 512, $lr = 0.001$ and $wd = 0.0001$. The maximum number of epoches is 1000. The reference distribution for our mTRE is taken to be the standard Gaussian distribution. Here, the reference distribution is the same as the noise distribution in the MNIST experiments of (Rhodes et al., 2020). We obtain the averaged ANLLs and their empirical standard errors for mTRE-5 and mTRE-10 over 5 random training subsamples. As a comparison, we use the results with the Gaussian noise for the direct single ratio estimate and the direct estimate based on the original convolution chain (cTRE) obtained from Table 1 in (Rhodes et al., 2020) as the benchmarks. Note that cTRE and the direct single ratio estimate are based on the full training sample, where the sample size is 60,000. The result for the cTRE presented in Table A.2 is the best one among the cTRE's with the chain length $B \in \{5, 10, 15, 20, 25, 30\}$ in Table 1 in the online supplemental of Rhodes et al. (2020). Our results are presented in Table A.2.

From Table A.2, we see that mTRE is significantly better than the single ratio estimate and comparable with cTRE. The difference between the results from mTRE and cTRE is not statistically significant. We note that the training sample size for mTRE we used is restricted to 20,000, due to the memory limitation of the laptop used in the computation. In comparison, the sample size for cTRE is 60,000.

Table A.1: Architecture for mTRE

| Layers | Details | Output size |
|---|---|---|
| Convolution | $3 \times 3$ Conv | $12 \times 28 \times 28$ |
| Transition Layer 1 | BN, ReLU, $2 \times 2$ Average Pool,$1 \times 1$ Conv | $12 \times 14 \times 14$ |
| Dense Block 1 | BN, $1 \times 1$ Conv, BN, $3 \times 3$ Conv | $24 \times 14 \times 14$ |
| Transition Layer 1 | BN, ReLU, $2 \times 2$ Average Pool,$1 \times 1$ Conv | $12 \times 7 \times 7$ |
| Dense Block 1 | BN, $1 \times 1$ Conv, BN, $3 \times 3$ Conv | $24 \times 7 \times 7$ |
| Pooling | BN, ReLU, $7 \times 7$ Average Pool, Reshape | 24 |
| Fully connected | Linear | 1 |

Table A.2: Average negative log-likelihood (ANLL) in bits per dimension (bpd, smaller is better). For the proposed mixing chain estimate with the chain length $B$ (mTRE-$B$), the ANLLs are averaged over 5 random training subsamples, where the subsample size is 20,000, and the corresponding standard errors are in parentheses. The cTRE is the direct estimate based on the original convolution chain (cTRE). The results for the direct single ratio estimate and the direct cTRE are obtained from Table 1 in the seminal paper (Rhodes et al., 2020) and we use them as the benchmarks. The cTRE and the direct single ratio estimate are based on the full training sample, where the sample size is 60,000.

| Estimator | mTRE-5 | mTRE-10 | Direct Single ratio | Direct cTRE |
|---|---|---|---|---|
| ANLL | 1.40 (0.0045) | 1.39 (0.0077) | 1.96 | 1.39 |

