# OpenReview forum: "An Error Analysis of Deep Density-Ratio Estimation with Bregman Divergence"
_NeurIPS.cc/2022/Conference — NeurIPS 2022 Submitted_

### Official Review · Reviewer_T57B · 2022-07-01

**Rating:** 7
**Confidence:** 4
**Soundness:** 3 good
**Presentation:** 3 good
**Contribution:** 3 good

**Summary:**

In this paper, the authors aim to establish non-asymptotic error bounds for a nonparametric density-ratio estimator which minimizes the empirical Bregman Divergence (BD) score over a class of neural network functions. More specifically, let $D^*=\log R^*$ where $R^*=p^*/q^*$ is the density-ratio that we would like to estimate. In addition, assume that we have $n_p$ i.i.d. samples from $p^*$ and $n_q$ i.i.d. samples from $q^*$.  The empirical nonparametric density-ratio estimator of $D^*$ over a class of feed-forward neural network function $\mathcal{F_{\mathrm{FNN}}}$ is defined as $\hat{D}:= argmin_{D \in \mathcal{F_{\mathrm{FNN}}}} \hat{\mathcal{B_{\psi}}}(e^D)$ where $\hat{\mathcal{B_{\psi}}}(\cdot)$ is the empirical Bregman Divergence (BD) score.

The authors first show that under some mild conditions, with high probability, $||\hat{D}-D^*||$ is bounded above by $||D_{\mathrm{NN}}-D^*||$ plus some $O(\sqrt{(\log n) \mathrm{Pdim}(\mathcal{F_{\mathrm{FNN}})}/n)}$ terms where $D_{\mathrm{NN}}= argmin_{D\in \in \mathcal{F_{\mathrm{FNN}}}} ||D-D^*||$, $\mathrm{Pdim}(\mathcal{F_{\rm{FNN}}})$ is the pseudo-dimension of $\mathcal{F_{\mathrm{FNN}}}$, and  $n:=\min (n_p,n_q)$ (a PAC-bound). Then, in the next parts, the authors show how to bound $||D_{\mathrm{NN}}-D^*||$ for a special case where $D^*$ belongs to a Holder class of functions, $\mathcal{H^{\beta}}([0,1]^d,M)$. The authors proved that this term is bounded by a term of order $(KL)^{-2\beta/d}$ if the neural network has with $\mathcal{W}=38(\lfloor \beta \rfloor+1)^2 d^{\lfloor \beta \rfloor +1} L\lceil \log_2(8L)\rceil$ and $\mathcal{D}=21(\lfloor \beta \rfloor+1)^2 K \lceil \log_2(8K)\rceil$. By additional assumption that the data from $p^*$ and $q^*$ are concentrated on some $\rho$-cover of a compact finite-dimensional Riemannian submanifold in $[0,1]^d$, the authors can obtain an improved bound on  $||D_{\mathrm{NN}}-D^*||$ which reduces the curse of dimensionality. Finally, the authors perform error analysis for a telescoping density-ratio estimator and affirm that the telescoping strategy (TRE) can improve the expected error $||\hat{D}-D^*||$ as $n\to \infty$, which affirms a conclusion by Rhodes et al. (2020) by experiments.

The key contribution of this paper is Theorem 1 which provides a PAC-bound for $||\hat{D}-D^*||$ via $||D_{\mathrm{NN}}-D^*||$.  The proof of this theorem is based on a good observation that the Bregman divergence score function $\hat{\mathcal{B_{\psi}}}(t)$ is sum of two Lipschitz functions and an interesting fact that $\hat{\mathcal{B_{\psi}}}(e^D)- \hat{\mathcal{B_{\psi}}}(e^{D^*})\sim ||D-D^*||^2$ under some bounded conditions. Hence, the authors can apply some classical results in learning theory such as symmetrization inequalities and Talagrand's contraction theorems to bounds $||\hat{D}-D^*||$. The other results are mainly based on existing results related to an approximation of a Holder function by a neural network function (Jiao et al. (2021)) and/or another dimension reduction result by (Nakada and Imaizumi, 2020) to bound $||D_{\mathrm{NN}}-D^*||$.

**Questions:**

Here is a list of typos and questions:

1. In line 586, please explain how to bound $||\tilde{D_E^*}(Ax')-D^*(x)||$ by $M\rho$. It looks not clear how to obtain this.
2.  Please explain why the convergence rate in (8) is optimal (a conclusion in line 166). See my comments in the previous part.
3.  Some typos: in line 134; Proposition 1 ($2 M_0$ should be replaced for $M_0$); a definition of $\Delta_{\psi}(x,y):=\psi(x)-\psi(y)-\psi'(x) (y-x)$ should be mentioned in the proof of Lemma A.2; a citation for (A.12) should be provided; and the norm of $\rm{SL}$ (not of its derivative) in line 569, etc. Please correct them.


**Limitations:**

This is a theoretical work. The authors already finished the checklist as well as stated the limitations of theorems and results.

**Strengths And Weaknesses:**

This work can be considered a novel combination of well-known techniques. The error bounds in this paper are new (to my knowledge). By setting a distribution is fixed, the error bound results in this paper can recover the minimax rate of the density estimation in Tsybakov (2008) with an improvement on the effects of $d$ (cf. Section 3.2).

In general, all the results in this paper are well proved and look correct. There are some typos but it does not hinder readers from understanding the results and their proofs. There is a problem related to the minimax optimal conclusion in line 168. More specifically, the authors mentioned that the convergence rate is optimal. This looks not make sense if the authors only based on the fact that this bound is minimax optimal when fixing a distribution. Optimality by fixing one does not imply optimality for the fraction of the two distributions.

From the technical viewpoint, there are some interesting ideas in the proofs such as converting the estimation problem for $R^*$ to the estimation problem for $D^*=\log R^*$, partitioning the range of $\hat{D}$ to make use the fact of that if $||D-D^*||\leq r$ then $||D-D^*||_{n_p,n_q}\leq 2r$ to bound $||\hat{D}-D^*||$ in the proof of Theorem 1, or using i.i.d. Bernoulli random variables to telescope density-ratio estimator in Section 4.

However, this paper contains some weak points as follows.
   1. To obtain an upper bound on $||D_{\mathrm{NN}}-D^*||$, the authors need to make an assumption that $D^*$ belongs to some special class of functions. More specifically, the authors assumed that $D^* \in \mathcal{H^{\beta}}([0,1]^d,M)$. However, this looks unavoidable with deep-learning-based methods. An example of $p^*$ and $q^*$ (common distributions, exponential class for example) such that $R^*$ belongs to this class will make your paper more interesting.
  2.  If $D^*$ is unbounded above, the result in Theorem 2 looks not tight enough.
  3. The optimality of the bounds in Theorem 3  needs to justify (see the comment below).

---

> ### Author Response · Authors · 2022-08-02
> **Response to Reviewer T57B**
>
> We are very grateful to you for your taking the time to read our paper and for your helpful comments and constructive suggestions.
> We also appreciate your noticing the merits of our work.
>
> ``To obtain an upper bound on $\vert D_{NN}-D^{\*}\vert$,  the authors need to make an assumption that $D^*$ belongs to some special class of functions. More specifically, the authors assumed that  $D^{\*}\in \mathcal{H}^\beta([0,1]^d,M).$   However, this looks unavoidable with deep-learning-based methods. An example of $p^*$ and $q^*$ (common distributions, exponential class for example) such that $R^*$ belongs to this class will make your paper more interesting."
>
>
> Response: Thank you for this suggestion. Indeed, the exponential class satisfies the requirement.
>
> $Exp(\beta,B)=\\{p(z)=\exp(g(z))/c_{g}:z\in [0,1]^d,c_{g}=\int_{[0,1]^d}\exp(g(t))dt,\ \ g\in\mathcal{H}^\beta([0,1]^d,B)\\},
> $
>
> if $p^*,q^* \in Exp (\beta,B)$, the logarithm of the density ratio $D^*(z)=\log (q^*(z)/p^*(z))$ also belongs to the H\"older class.
> We have added this example and a more specific example in the appendix in the rebuttal revision.
>
> "If $\|D^*\|$ is unbounded above, the result in Theorem 2 looks not tight enough."
>
> Response: To the best of our knowledge, the tight bound for the density-ratio estimate in unbounded setting is an open question.  Even in density estimation problems, when the density is unbounded above, it remains unsolved what the tight bound is.  Our result in Theorem 2, albeit not tight, is an attempt to tackle the challenging unbounded density-ratio problem.
>
> "The optimality of the bounds in Theorem 3 needs to justify (see the comment below)."
>
> Response: The optimality can be justified as follows. If our obtained convergence rate is not minimax optimal, then there exists a convergence rate faster than our result. That means, for the target density ratio $D^{\*}\in\mathcal{H}^\beta([0,1]^d,M)$, there exists an estimate $\hat{D}$ of $D$, which is based on samples $(Z\_{q,i})\_{i=1}^{n\_q}$  from  ${q^{\*}}$  and $(Z\_{p,j})_{j=1}^{n\_p}$ from $p^{\*}$,  such that
>
> $E\_{p^*, q^*}\vert \widehat{D}-D^*\vert_{\max}^2\le O(n^{-\xi}),\ \xi>  \frac{2\beta}{d+2\beta},\quad (1) $
>
> where $n=\min (n_q,n_p)$.
>
> Now considering a density estimation problem with i.i.d observations $(Z\_{q,i}^{(1)})\_{i=1}^{m_q}$ from an underlying unknown density $q\_1$ on $[0,1]^d.$  To estimate $q\_1$, we sample referencing observations $(Z_{p,j}^{(1)})_{j=1}^{m_p}$ with $m_p\ge m_q$, from a uniform distribution Unif$([0,1]^d)$ whose density $p_1\equiv1$.  Thus, estimating the density ratio $q_1/p_1$ is equivalent to estimating $q_1$.  Using the density ratio estimate $\hat{D}$ of $D$, we obtain the estimator $\hat{q}_1$ of $q_1$. If $\log q_1\in\mathcal{H}^\beta([0,1]^d,M)$
> where $\beta=k+a$ with $k\in \mathbb{N}^+$ and $a\in(0,1]$, then by the above inequality (1), $\hat{q}_1$ satisfies
>
> $E_{p_1,q_1}\|\hat q_1-q_1\|_{max}^2\le O(m_q^{-\xi}),\ \xi > \frac{2\beta}{d+2\beta}.$
>
> This result contradicts the classic optimal minimax convergence rate $O_p\left(m_q^{-{2\beta}/{(d+2\beta)}}\right)$ for the density estimation when a density belonging to the H\"older function class.  Hence, our estimator achieves the optimal minimax rate.
>
> "In line 586, please explain how to bound
> $|\tilde{D}^*_E(Ax')-D^*(x)|$ by $M\rho$. It looks not clear how to obtain this.''
>
> Response: It follows from the definition of $\tilde{D}^*_E$ that $\tilde{D}^*_E\equiv\tilde{D}^*$ on $A(\mathcal{M})$.  And by the definition of $\tilde{D}^*$, we have $\tilde{D}^*(At)=D^*(t)$ for any $t\in\mathcal{M}$.
> Then,  for
> $x'\in\mathcal{M}$ and $x\in\mathcal{M}_\rho$ with $\|x'-x\|_2\le\rho$, we have
>
> $$|\tilde{D}^*_E(Ax')-D^*(x)|=|D^*(x')-D^*(x)| \le M\|x'-x\|_2\le M\rho,$$
>
> where the first inequality follows from the fact that $D^*\in\mathcal{H}^\beta([0,1]^d,M)$.
>
> "Some typos: in line 134; Proposition 1 ($2M_0$ should be replaced for $M_0$); a definition of $\Delta_{\psi}(x,y):=\psi(x)-\psi(y)-\psi'(x)(x-y)$ should be mentioned in the proof of Lemma A.2; a citation for (A.12) should be provided; and the norm of $\mathcal{SL}$ (not of its derivative) in line 569, etc. Please correct them."
>
> Response:   Thank you for your careful reading of our paper.  In the revision,
> we have added the missing reference Section 4 in line 134; regarding  the prefactor in Proposition 1,
> it should be  $e^{M_0}$ instead of $e^{2M_0}$, as the bound in Proposition 1 is for the norm
> $\vert\widehat{D}-D^*\vert\_2$, but not its squared version $\vert\widehat{D}-D^{\*}\vert\_2^{2}$; we have added the definition of $\Delta_{\psi}(x,y):=\psi(x)-\psi(y)-\psi'(x)(x-y)$ in the proof of Lemma A.2 in the revision; a citation for (A.12) is now provided in line 535; and we have revised the relevant statement about $\mathcal{SL}$ now in line 599 of the revision.

---

> > ### Comment · Reviewer_T57B · 2022-08-05
> > **Reply to the rebuttal**
> >
> > The authors had answers to all my comments satisfactorily, especially added an example of $p^*$ and $q^*$ such that the ratio belongs to the H\"{o}lder class. Hence, I raised my score to accept.

---

> > > ### Author Response · Authors · 2022-08-05
> > > **Reply to the additional comment**
> > >
> > > We are very happy to know that you were satisfied with our response to your comments.
> > > Thank you so much for raising your score to accept!

---

### Official Review · Reviewer_Eu9B · 2022-07-06

**Rating:** 6
**Confidence:** 3
**Soundness:** 2 fair
**Presentation:** 2 fair
**Contribution:** 3 good

**Summary:**

This paper establishes non-asymptotic error bounds for a density-ratio estimator based on the Bregman divergence. These bounds depend on the approximation error and pseudo-dimension of the underlying function classes. If one assumes the optimal estimator belongs to the Hölder class, combing with existing results on the approximation error of neural networks to Hölder functions, and the pseudo-dimension of neural networks, the authors establish error bounds of neural-network based density-ratio estimators. If the data is concentrated on a low-dimensional manifold, the curse of dimensionality can be alleviated. The authors also apply the error bounds to analyze a telescoping density-ratio estimator.

**Questions:**

1. Do you have any evidence that the chain defined in the paper has a better or similar performance to the one in [2]?
2. Can the convergence result in [1] induce similar result on the chain problem defined in the paper or the one in [2], i.e., the chains are better than a single-ratio estimator?

**Limitations:**

No other major concerns than the ones listed in weaknesses.

**Strengths And Weaknesses:**

Strengths:
This paper provides a general non-asymptotic bound for the density-ratio estimator, which does not depend on any specific function class. With known neural-network theoretical properties, this general bound can be applied to the deep density-ratio estimator, which is in some sense optimal. The convergence rate is also slightly faster than known result in [1].

Weaknesses:
1. The paper is not carefully written. Line 134 contains a missing reference. Line 260 uses an abbreviation TDR (I believe it is a typo of TRE?), which is not defined anywhere.
2. The authors analyze a different telescoping density-ratio estimator, rather than the one in [2], and acknowledge that the techniques used for the chain defined in the paper do not apply to the estimator of interest in [2]. This means that the analysis is only for the problem defined in this paper. The authors should provide some evidence that the chain defined in this paper has a superior or comparable performance to the estimator of interest in [2].

[1]Kato, M. and Teshima, T. Non-negative bregman divergence minimization for deep direct density ratio estimation. In International Conference on Machine Learning, pp. 5320 – 5333, 2021.

[2] Rhodes, B., Xu, K., and Gutmann, M. U. Telescoping density-ratio estimation. In Advances in Neural Information Processing Systems, 2020.

---

> ### Author Response · Authors · 2022-08-02
> **Response to Reviewer Reviewer Eu9B**
>
> We are very grateful to you for your taking the time to read our paper and for your helpful comments and constructive suggestions.
>
> ``Strengths: This paper provides a general non-asymptotic bound for the density-ratio estimator, which does not depend on any specific function class. With known neural-network theoretical properties, this general bound can be applied to the deep density-ratio estimator, which is in some sense optimal. The convergence rate is also slightly faster than known result in [1]."
>
> Response: Thank you very much for noticing the merits of our paper.
>
> ``The paper is not carefully written. Line 134 contains a missing reference. Line 260 uses an abbreviation TDR (I believe it is a typo of TRE?), which is not defined anywhere."
>
> Response: We have added the missing reference and change "TDR" to "TRE" in the revision. We shall improve the exposition  and remove possible typos in the revision.
>
> "The authors analyze a different telescoping density-ratio estimator, rather than the one in [2], and acknowledge that the techniques used for the chain defined in the paper do not apply to the estimator of interest in [2]. This means that the analysis is only for the problem defined in this paper. The authors should provide some evidence that the chain defined in this paper has a superior or comparable performance to the estimator of interest in [2]."
>
> Response: A major difficulty in analyzing the original chain is the intensive oscillation of the density ratio incurred by the convolution form of two density functions, which leads to unstable and unbounded density ratios. Our proposed mixing chain can overcome the theoretical difficulty of the convolution chain and is easier to analyze. Our additional simulation studies contain evidences that our proposed mixing chain indeed can have  better performance compared with the original convolution chain in some cases; see our response to your question below for more numerical results.
>
> ``Do you have any evidence that the chain defined in the paper has a better or similar performance to the one in [2]?''
>
> Response: Yes,  we have conducted additional simulations to verify empirically the performance of our proposed mixing chain and the original convolution chain in [2]; see Table 2 in the revision.  The simulation settings are given on page 25 in the Appendix of  the rebuttal revision.  The results show that, for the models considered in the simulation studies, the proposed mixing chain performs comparably or better compared with the original convolution chain.
>
> ``Can the convergence result in [1] induce similar result on the chain problem defined in the paper or the one in [2], i.e., the chains are better than a single-ratio estimator?"
>
> Response: Thanks for the comment. The convergence result in [1] can not induce similar result  as in Proposition 1 of our  paper.  Our theoretical analysis takes the approximation error into account, while the convergence result in [1] relies on the assumption that the target ratio belongs to the optimization space (or hypothesis space) under which there is no approximation error.
> To apply the convergence result in [1] for the telescopic ratio estimate (TRE), one needs to assume that all density ratios in the chain belong to the optimization space. This means that all the density ratios in the chain themselves are
> ReLU neural networks, which is not satisfied here.
>
>
> [1]Kato, M. and Teshima, T. Non-negative bregman divergence minimization for deep direct density ratio estimation. In International Conference on Machine Learning, pp. 5320 – 5333, 2021.
>
> [2] Rhodes, B., Xu, K., and Gutmann, M. U. Telescoping density-ratio estimation. In Advances in Neural Information Processing Systems, 2020.

---

> > ### Comment · Reviewer_Eu9B · 2022-08-03
> > **Question on the empirical evaluation**
> >
> > I notice that the original paper of [2] used MNIST, while the provided empirical evaluation only considered synthetic data. What is the difficulty of using the standard MNIST data for evaluation?

---

> > > ### Author Response · Authors · 2022-08-05
> > > **Response on the question on the empirical evaluation**
> > >
> > > We have only added the empirical evaluation using synthetic data because it is faster to run so that we could
> > > include the results in our response before the deadline. We have started to run our method on the MNIST data. It will take longer than with the synthetic data to get the results,  since the MNIST dataset is bigger than the synthetic dataset we generated and there is the need to tune many networks for the density ratios in the chain.
> > >
> > > We note that in the appendix of [2], the authors also commented on the long time needed in their experiments with the MNIST dataset, in particular when a long chain is used. We will include the results on the MNIST data in the next revision and will update you as soon as we have the results. Thank you very much and we appreciate your question and suggestion.

---

> > > > ### Comment · Reviewer_Eu9B · 2022-08-06
> > > > **post-rebuttal response**
> > > >
> > > > I have read the reviews and responses. While it is still not clear what the empirical result is on the nonstandard data, the authors have addressed my concerns, so I would like to increase my score.

---

> > > > > ### Author Response · Authors · 2022-08-08
> > > > > **Added MNIST data results**
> > > > >
> > > > > First, we are happy to know that we have addressed your concerns and thank you so much for increasing your score, we really appreciate it!
> > > > >
> > > > > Just to update that we have completed some preliminary analysis of the MNIST.
> > > > > Our results are similar to the result based on the original chain in [2].
> > > > > In particular, the average negative log-likelihood in bits per dimension calculated based on our proposed chain is similar to that from [2]. The differences are not statistically significant. We have added the detailed description in Appendix A.5 in the updated rebuttal revision. We will conduct more experiments to evaluate our method.
> > > > >
> > > > > Again, we are very grateful to you for your review and helping us improve our paper.

---

### Official Review · Reviewer_hTC4 · 2022-07-11

**Rating:** 6
**Confidence:** 3
**Soundness:** 3 good
**Presentation:** 1 poor
**Contribution:** 3 good

**Summary:**

This paper studies density ratio estimation based on deep ReLU networks. Based on a loss criterion by the Bregman divergence (eq.1), and under weak conditions (stated in assumptions 1-3), the authors showed that this family of density ratio estimators can achieve optimal convergence rate (Thm 1 & 2, and 3).

The authors also extended their bound into the cases where the data is on a low-dimensional submanifold (Thm 4), showing the curse of dimensionality can be alleviated.

As an application, the authors further applied their bounds to the Telescoping density-Ratio estimation (TRE) (Rhodes et al., 2020). The author's bounds can help demonstrate the advantages of TRE over single-ratio estimators.

**Questions:**

The current paper studied density estimation of p and q, which are pdfs supported on a hypercube [0,1]^d. It is not clear how the results can be generalized.

Assumption 2. As a ReLU network output values in the range (0, \infty). It is not clear how this assumption can be satisfied. It should be explained in more detail in the text.

In equation (3),  $\hat{D}$ is the global optimum by definition. In practice, one only gets a local optimum solution through neural network training. Can the stated bounds guarantee the density-ratio estimation quality of these local optimums?

The main weaknesses mentioned in "Strengths And Weaknesses" can be regarded as my questions.

**Limitations:**

This work is of theoretical nature. I could not see any negative societal impact.

**Strengths And Weaknesses:**

Pro:
- Solid theoretical results.

Con:
-The paper contains a list of formal results and lacks intuitions, numerical examples, and detailed explanations. As a result, it is hard to read.
- The theoretical results are for general Bregman divergences satisfying certain smoothness and convexity constraints. Although some examples are given in table 1, it is not clear how certain choices of the Bregman divergence can further enhance the results.

Please see below for some minor comments (these are not questions to be addressed in the rebuttal).

eq.(1), write some intermediate step, as the reader may be familiar with Bregman divergences but not eq.(1) and wonder where it comes from.

the unnumbered equations between eq.(1) and eq.(2): $R\in\cdots$ (use set notations)

L134: Section ?? below (missing reference).

Pdim: mention the relation with VC dimension (that is widely known).

---

> ### Author Response · Authors · 2022-08-02
> **Response to Reviewer hTC4**
>
> We are very grateful to you for your taking the time to read our paper and for your helpful comments and constructive suggestions.
>
> ``Pro: Solid theoretical results.''
>
>   Response: Thank you very much for noticing the merits of our paper.
>
> ``Con: The paper contains a list of formal results and lacks intuitions, numerical examples, and detailed explanations. As a result, it is hard to read."
>
> Response: We have now added a high-level description of the main ideas of the proof.   We decompose the excess risk into two parts:  statistical error and approximation error.  To bound the statistical estimation error, we apply the empirical process theories;  for the approximation error, we employ function approximation results using deep neural networks.
>
> In the revision, we have also added numerical examples to compare the performance of our proposed mixing chain and the original convolution chain; see Table 2 in the revision for the new results.  The results show that, for the models considered in the simulation studies, the proposed mixing chain performs comparably or better compared with the original convolution chain.
>
> ``... it is not clear how certain choices of the Bregman divergence can further enhance the results."
>
> Response: It is difficult to state unequivocally certain choices of the Bregman divergence will lead to better results. Our empirical experience with some specific forms of the Bregman divergence, including the least squares loss, the logistic loss and the KL loss, indicate that they have similar performance, with the logistic loss performs slightly better in some cases.
>
> ``Pdim: mention the relation with VC dimension (that is widely known)."
>
> Response: We have added the following remark in the revision.
>
> For any measurable function class $\mathcal{F}$, by the definition of VC dimension,  $\textup{VCdim}(\mathcal{F})\le\textup{Pdim}(\mathcal{F})$. If $\mathcal{F}$ is the class of functions generated by ReLU FNNs, by Theorem 14.1 of Anthony and Bartlett (1999), it holds $\textup{Pdim}(\mathcal{F})\le\textup{VCdim}(\mathcal{F})$. Hence, for the function class $\mathcal{F}$ generated by ReLU FNNs, $\textup{Pdim}(\mathcal{F})=\textup{VCdim}(\mathcal{F})$.
>
>
> ``The current paper studied density estimation of p and q, which are pdfs supported on a hypercube $[0,1]^d$. It is not clear how the results can be generalized."
>
> Response: With unbounded support, we can bound  $\vert D_{\text{NN}}-D^*\vert_{\text{max}}$ using a truncation technique under some additional assumptions, at a small price of an additional logarithm term in the error bound. We have added the following result, details are included in the appendix in the revision.
>
> When the pdfs are supported on ${R}^d$, in addition to Assumptions 1-2 and the H\"older assumption in Theorem 3, we need to  further assume the following tail probability condition: for $a_n = E_{p^*}I(\vert Z\vert_\infty \ge\log n)$ and  $b_n = E_{q^*}I(\vert Z\vert_\infty\ge\log n)$, it holds that $\max (a_n, b_n) \le n^{-{2\beta}/{(d+2\beta)}}$. With this additional condition, we can show that
> $\Vert D_{NN}-D^*\Vert_{max}^2\le328M^2(\lfloor\beta \rfloor+1)^4d^{2\lfloor\beta \rfloor+(\beta\vee 1)}(2\log n)^{2\lfloor\beta \rfloor}n^{-\frac{2\beta}{d+2\beta}}.$
>
> Compared with the bound in the bounded support case in Theorem 3,  this bound has an additional logarithmic factor $(2\log n)^{2\lfloor\beta \rfloor}$. Details are given in the appendix in the revision.
>
> ``Assumption 2. As a ReLU network output values in the range (0, $\infty$). It is not clear how this assumption can be satisfied."
>
> Response: Assumption 2 is a common technical assumption. Similar assumptions can be found  in [2].  The boundedness can be achieved by clipping operation.  For example, let $T_M(t)=-MI(t\textless -M)+tI(-M\le t \le M)+MI(t\textgreater M)$ be the truncation function taking values in $[-M,M]$, where $I(\cdot)$ is the indicator function, then $T_M(t)=\sigma(t)-\sigma(\sigma(t)-M)-\{\sigma(-t)-\sigma(\sigma(-t)-M)\}$ can be computed by a ReLU network with depth $2$ and width $4$.  Hence, through network concatenation, we can construct bounded ReLU FNNs.  We have added explanations on this in the revision.
>
>
> ``In equation (3), $\hat{D}$ is the global optimum.... Can the stated bounds guarantee the density-ratio estimation quality of these local optimums?"
>
> Response:  In the empirical risk minimization, the global minimizer is the focus of the theoretical analysis. Our results may not hold for local solutions. To study the properties of local optima, one needs to analyze the optimization error. This is an interesting and challenging problem that deserves further study in the future.
>
> [1] Martin Anthony and Peter Bartlett. Neural network learning: theoretical foundations. Cambridge U Press, 1999.
>
> [2] Nakada, R. and Imaizumi, M. (2020). Adaptive approximation and estimation of deep neural network with intrinsic dimensionality. JMLR, 21, 1 - 38.

---

> > ### Comment · Reviewer_hTC4 · 2022-08-10
> > **Rebuttal Read**
> >
> > I would like to thank the authors for the detailed feedback. The authors revised the papers based on the reviews and enhanced the presentation. They also showed the results can be generalized to densities on $R^d$. I lean towards acceptance after reading the rebuttal.
> >
> > Note L59-L62, $E_p\Delta_\phi\ge0\leftrightarrow{R}=R^\star$ depends on the support of p where the density is greater than 0, which is not mentioned here (or before). If the support of p and q is not the full hypercube, the result may be different. Here, can eq.~(2) be introduced without referring to the appendix?
> >
> > L119-L122 please make sure all symbols are introduced: Iverson bracket, $\sigma$, $\dots$

---

> > > ### Author Response · Authors · 2022-08-10
> > > **Thank you for your response and additional comments**
> > >
> > > Thank you so much for your positive feedback to our response and revision.
> > > We also appreciate your additional comments and suggestions. We will made changes accordingly in the next revision.
> > >
> > > L59-L62: we have now changed to $E_{p^*}\Delta_{\phi}=0$ if and only if $R=R^*$ almost everywhere with respect to the probability distribution with density $p^*$.
> > >
> > > L119-122: we have defined the notations, including Iverson bracket and $\sigma$.
> > >
> > > We will provide the detailed derivation of Eq (2) in the main context, without referring to the appendix.
> > >
> > > Thank you so much again for all your work reviewing our paper and helping us improve it,
> > > we really appreciate your work and help!

---

### Official Review · Reviewer_1fM2 · 2022-07-12

**Rating:** 5
**Confidence:** 2
**Soundness:** 2 fair
**Presentation:** 2 fair
**Contribution:** 2 fair

**Summary:**

The paper introduces a density ratio estimator (DRE) based on the Bregman divergence which is different from Kato & Teshima (2021).
The author(s) derive(s) non-asymptotic error bounds for the proposed DRE, which turns out to be slightly better than that of Kato & Teshima (2021).
Based on the established bounds, the paper then studies why the telescoping density-ratio estimator (TRE) has a good empirical performance compared to single-ratio DRE.
On a specific bridging methods proposed by the author(s), it was argued that TRE has a better asymptotic error bound than that of the single-ratio method.
It also indicates that the originally proposed bridging method in TRE could have some insatiability issue.

**Questions:**

- $\exp$ is used to make the density ratio estimator positive but it is known that $\exp$ is numerically unstable and in deep learning usually $\mathrm{softplus}$ is preferred. Can you justify your choice here?

Typos
- L134: "Section ??" -> Section 4
- L260: "TDR" -> TRE

**Limitations:**

Limitations are discussed but not potential negative societal impact.
The work is theoretical and I don't see direct societal impact.


**Strengths And Weaknesses:**

Pros
- The paper studies an important problem that wasn't paid enough attention.
- The method and analysis presented in the paper is novel.

Cons
- Conclusions derived from the analysis are not verified empirically. For example the choice of bridging method (mixing v.s. convolution).
- The analysis of TRE is based on an alternative bridging method for analysis convenience, which may not be enough to explain why TRE performs well.
- The paper is dense and not easy to follow for readers who are not familiar with the proof techniques.

---

> ### Author Response · Authors · 2022-08-01
> **Response to Reviewer 1fM2**
>
> We are grateful to you for taking the time to review our paper and noticing the merits of our work.
>
>  ``Conclusions derived from the analysis are not verified empirically. For example the choice of bridging method (mixing v.s. convolution)."
>
> Response: Thank you for your comments. It is generally hard to verify the convergence rates empirically, as numerical optimization algorithms such as  SGD may not find the global optimal solution.  Regarding the choice of the bridging method, we have conducted additional simulations to evaluate the performance of our proposed mixing chain and the original convolution chain; see Table 2 in the revision for the new results.  The simulation settings are given on page 25 in the Appendix of the rebuttal revision.  The results show that, for the models considered in the simulation studies, the proposed chain performs similarly to or better than the original convolution chain.
>
> ``The analysis of TRE is based on an alternative bridging method for analysis convenience, which may not be enough to explain why TRE performs well."
>
> Response: A major difficulty in analyzing the original chain is the intensive oscillation of the density ratio incurred by the convolution form of two density functions, which leads to unstable and unbounded density ratios. Our proposed mixing chain can overcome the theoretical difficulty of the convolution chain and is easier to analyze.  Our additional simulation studies contain evidences that our proposed mixing chain indeed can have  better performance compared with the original convolution chain in some cases. It would be an interesting problem to apply or revise the techniques in our theoretical analysis of the mixing chain  to study the properties of general TRE.
>
> ``The paper is dense and not easy to follow for readers who are not familiar with the proof techniques."
>
> Response: Indeed, our paper is of theoretical nature and a little dense. We will improve the exposition in the main text and the appendix in the revision, to make the paper reader friendly.
>
> ``$\exp$ is used to make the density ratio estimator positive but it is known that $\exp$ is numerically unstable and in deep learning usually softplus is preferred. Can you justify your choice here?"
>
> Response: We wish to clarify that $R$ is the density-ratio in our paper and $D$ is defined as the logarithm of $R$, i.e., $D=\log R$.  After  obtaining   the estimator of  $D$, denoted by $\hat{D}$,  the estimator of $R$ is $\exp(\hat{D})$ by  the plug-in rule.   Hence, $\exp(\cdot)$ is involved here as it is the inverse of  the logarithmic function, but not to make the density ratio estimator positive.
>
> ``Typos: "Section ??" $\to$ Section 4. "TDR" $\to$ TRE.
>
> Response: It should be "Section 4". Changed  "TDR'' to  "TRE".

---

> > ### Author Response · Authors · 2022-08-07
> > **Comments/questions about our rebuttal?**
> >
> > Thank you very much again for your work reviewing our paper.
> > We are just wondering if you have any comments/questions about our rebuttal.
> > Please just let us know if there are any points that we have not explained clearly
> > in our rebuttal or if you have any further comments/questions.
> > We would be happy to provide additional clarifications.
> > Many thanks!

---

> > ### Comment · Reviewer_1fM2 · 2022-08-09
> > **Reply to author response**
> >
> > Re. "the choice of bridging method (mixing v.s. convolution)"
> >
> > Thanks for the extra results, which looks positive to me.
> > I raised my score accordingly following the paper update.
> >
> > I suggest to make it clear upfront that the paper's theory tells the choice of bridging methods is important (the oscillation point) and it is used to choosing the right bridging which improves TRE.
> > The current wording of Contribution 3 starting from L47 really makes the readers expect to see an analysis on TRE as it is presented in the original paper.
> > But what the paper does is analyzing the TRE as a framework with a different bridging method.
> > Maybe something like below would work better?
> > "We apply our results to study the convergence properties of the telescoping density-ratio estimator (Rhodes et al., 2020) with the mixing waymark creation and demonstrate its advantages over single-ratio estimator under certain conditions and over the originally proposed convolution waymark creation method."

---

> > > ### Author Response · Authors · 2022-08-09
> > > **Thank you for your additional comments**
> > >
> > > Thank you very much for your positive feedback about our update.
> > > We are very grateful to you for raising your score!
> > >
> > > We also appreciate your additional comments on our paper. Indeed, we agree it is better to
> > > clearly state that our analysis concerns a mixing type of chain, instead of the original chain based on convolution. Your comments and suggestions are very helpful and we will incorporate them into our next revision.
> > >
> > > Again, thank you so much for your taking the time to review our work and helping us improve our paper!

---

### Meta-Review · Area_Chair_cNtp · 2022-08-29

**Recommendation:** Reject
**Confidence:** Less certain

**Metareview:**

This paper establishes non-asymptotic error bounds for nonparametric density-ratio estimators using deep neural networks.
According to the reviews this is a borderline paper and some changes will be needed before publication. In particular, some reviewers found the paper to be too dense and not easy to follow. The paper lacks intuitions, detailed explanations, and numerical illustrations.


**Award:**

No

---

### Decision · Program_Chairs · 2022-09-14

Reject